# Parameter-efficient Tuning of Large-scale Multimodal Foundation Model

**Haixin Wang**[1,*], **Xinlong Yang**[1,*], **Jianlong Chang**[2,†], **Dian Jin**[3], **Jinan Sun**[1,†],
**Shikun Zhang**[1], **Xiao Luo**[1], **Qi Tian**[2]
[1]Peking University, [2]Huawei Cloud & AI, [3]University of Wisconsin-Madison
{wang.hx, xinlong.yang}@stu.pku.edu.cn, {jianlong.chang, tian.qi1}@huawei.com,
djin38@wisc.edu, {sjn, zhangsk, xiaoluo}@pku.edu.cn

## Abstract

Driven by the progress of large-scale pre-training, parameter-efficient transfer learning has gained immense popularity across different subfields of Artificial Intelligence. The core is to adapt the model to downstream tasks with only a small set of parameters. Recently, researchers have leveraged such proven techniques in multimodal tasks and achieve promising results. However, two critical issues remain unresolved: *how to further reduce the complexity with lightweight design* and *how to boost alignment between modalities under extremely low parameters.* In this paper, we propose **A** grace**fu**l **pr**ompt framew**o**rk for c**r**oss-modal tr**a**nsfer (**Aurora**) to overcome these challenges. Considering the redundancy in existing architectures, we first utilize the mode approximation to generate 0.1M trainable parameters to implement the multimodal parameter-efficient tuning, which explores the low intrinsic dimension with only 0.04% parameters of the pre-trained model. Then, for better modality alignment, we propose the Informative Context Enhancement and Gated Query Transformation module under extremely few parameters scenes. A thorough evaluation on six cross-modal benchmarks shows that it not only outperforms the state-of-the-art but even outperforms the full fine-tuning approach. Our code is available at: `https://github.com/WillDreamer/Aurora`.

## 1 Introduction

The era of large models in deep learning has arrived, with increasingly more large-scale pre-trained models exhibiting remarkable generation and inference capabilities in the fields of text [38, 9, 63], vision [57, 53], and multi-modality [56, 41, 3, 76]. Nowadays, the mainstream strategy in the community involves pre-training models on large-scale data, and then fine-tuning them for each downstream task. This approach has been demonstrated to be effective in various cross-modal tasks, as evidenced by [40, 50, 67].

However, this high-performing strategy is not always advantageous, as several practical factors limit its further development. The primary concern is excessive parameter dependency. For example, GPT-3 [9] (175B params) obtains striking performances while coming at the cost of a massive parameter count. The enormous parameter size has two obvious drawbacks. Firstly, it incurs significant computations and physical storage costs, making pre-training and transfer very expensive. Besides, fine-tuning limits pre-trained knowledge's effectiveness in small-scale downstream tasks. Both of them hinder the further extension of large models from specific datasets to more general scenarios.

To relieve the high cost of large pre-trained models, a series of parameter-efficient transfer learning (PETL) [17, 74] methods have been proposed. The common paradigm is to freeze the backbone

---

*Equal contribution. †Corresponding author.

network of the large model and introduce a small number of additional parameters. Recently, some works have begun to focus on PETL in the multimodal field, such as UniAdapter [49], VL-Adapter [62], and MAPLE [33]. However, their common idea is to combine existing architectures used in NLP for multimodal models, which simply inserts learnable parameters in the backbone networks of unimodal and multimodal branches to achieve good performances. Their simple designs cannot integrate the essence of efficient parameter transfer into multimodal models. There are still two main challenges that need to face: one is *how to transfer knowledge in an extremely lightweight manner*, and the other is *how to boost alignment between modalities under extremely low parameters.*

To overcome the challenges mentioned above, we propose **A** grace**fu**l **pr**ompt framew**o**rk for c**r**oss-modal tr**a**nsfer, named **Aurora**, which surpasses the state of the art with a remarkably small parameter budget of merely ∼100K, thereby aptly living up to its name due to lightness. **First**, inspired by the idea of CANDECOMP/PARAFAC (CP) decomposition [64], we utilize mode approximation to generate lightweight learnable parameters for the attention-based architectures. As attention is proven to inevitably fit the background noise [45], most other features are redundant. For example, the categories in CLIP [56] are essentially infinite, which results in only a few features responding to a single class. This can be easily proven by the fact that the logits output by CLIP are mostly between 0.2 to 0.3 with very little fluctuation, indicating that most features are consistent. In other words, although pre-trained models have large-scale high-dimensional parameters, the intrinsic dimension corresponding to each downstream task is not large [55]. Thus, our mode approximation can fine-tune a very small number of parameters to achieve good performance on downstream tasks theoretically.

**Second**, in contrast to existing methods that directly insert a small network for cross-modal fusion, we mine deeper into the unimodal information and fuse them in a controllable manner with multimodal representations, thereby achieving desirable results with lower parameter dependence. In our proposed Informative Context Enhancement module, each feature is enriched by contextual information to boost the modality fusion. Furthermore, we adaptively control the ratio of textual information supplied for modality fusion in our proposed Gated Query Transformation module.

Finally, we evaluate Aurora on six cross-modal tasks and two zero-shot tasks, which obtains state-of-the-art performance compared to other PETL methods. Compared to the full fine-tuning, Aurora even obtains 1.8% and 0.5% performance improvement on MSRVTT and VQAv2 benchmarks averagely but only with about 0.05% trainable parameters of the pre-trained model.

## 2    Related Work

### 2.1    Vision-Language Models

In vision-language pre-training, large-scale image-text pairs are used to pre-train the Vision-Language Models (VLMs) for downstream tasks [13, 56, 41, 72, 3, 76]. The architecture of VLMs typically consists of a visual encoder, a textual encoder, and a cross-modal fusion module. VLMs have been applied to various tasks, including image generation [71], image captioning [39, 26], visual question answering [12, 11, 20], and cross-modal retrieval [58, 28]. This article focuses on how to efficiently fine-tune VLMs on downstream tasks and utilizes BLIP-base [41] for vision-language tasks.

### 2.2    Parameter-efficient Transfer Learning

As the size of recent models increases rapidly, updating the models in parameter-efficient ways becomes crucial. The huge computational burden brought by these pre-trained models will undoubtedly impose unnecessary burdens on transfer learning. PETL [17, 74] methods diverge from the conventional approach of fine-tuning the entire pre-trained model, instead only learning a few additional parameters for knowledge transfer. There are three common categories, prompt tuning [77, 30, 66, 46], adapter tuning [32, 10, 62] and parameter tuning [25, 31, 61]. We aim to relieve the computational burden by developing a graceful prompt framework specifically for cross-modal transfer, adapting frozen pre-trained multimodal models to downstream tasks across a broad distribution.

### 2.3    Tensor Decomposition

Tensor decomposition [35, 4, 59, 51] is an important research area studied for decades, which aims to approximate a tensor through a set of low-rank factors with diverse applications. In general, the

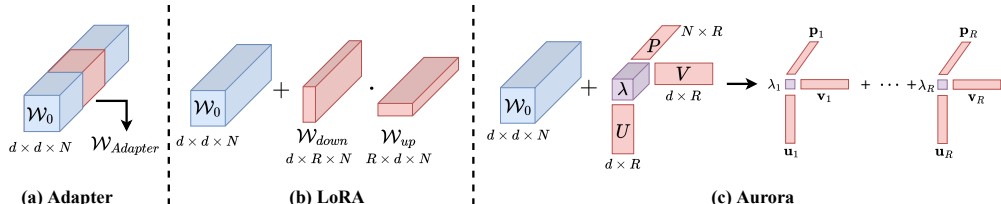

Figure 1: Comparison of existing PETL methods for downstream cross-modal tasks. (a) **Adapter**, which involves inserting a learnable small network into a pre-trained model; (b) **LoRA**, which employs a down and up tensor as updated parameters for low-rank approximation ($R \ll d$), added to the pre-trained model; and (c) our proposed **Aurora**, which utilizes mode approximation to further reduce the number of trainable parameters added to the pre-trained model. Notably, the red blocks represent trainable parameters, while the blue ones indicate the frozen backbone.

most widely used decompositions are Tucker decomposition [15] and CANDECOMP/PARAFAC (CP) decomposition [64], both of which can be seen as generalizations of the matrix singular value decomposition (SVD) [60]. CP decomposition can also be seen as a special Tucker decomposition whose core tensor is diagonal, but is more *lightweight* and *explainable*, meaning that a high-order tensor can be uniquely represented as the sum of a set of rank-one tensors theoretically. Here, we leverage the idea of CP decomposition to implement mode approximation for lightweight prompts.

## 3 Methodology

### 3.1 Background

**Frozen Backbone.** BLIP [41] is a unified VLP framework which has multimodal mixture of encoder-decoder(MED) architecture with both understanding and generation capabilities. We utilize BLIP-base as the frozen backbone and the pre-trained weights can be downloaded from Salesforce. Its visual encoder is ViT-B [19] and the text encoder is the same as BERT [16] while the text decoder replaces the self-attention layers with causal self-attention layers. It uses cross-attention layers to gather information from encoded visual representations using the textual representations as query. It's flexible to choose different components in the BLIP architecture to perform different multimodality downstream tasks.

We start with the core attention-based Transformer architecture mostly utilized in existing large-scale multimodal models [56, 44, 73, 41] for representation. The input image/text is divided into several non-overlapping patches, and then these patches appended a [CLS] token are fed into an embedding layer followed by the Transformer blocks with multi-head attention as the core operation. It copies the input embedding into query, key, and value by three projection matrices $W_q^l$, $W_k^l$, and $W_v^l \in \mathbb{R}^{d \times d}$ respectively, where $l$ represents the $l$-th layer. The pre-trained model contains many dense layers which perform matrix multiplication as follows:

$$\text{Attention}(X_{q,m}^l, X_{k,m}^l, X_{v,m}^l) = \text{Softmax}\left(\frac{X_{q,m}^l W_{q,m}^l \left(X_{k,m}^l W_{k,m}^l\right)^T}{\sqrt{d}}\right) X_{v,m}^l W_{v,m}^l$$

where $m \in \{I, T, C\}$ denotes the vision, text, and cross-modal modality. For the unimodal vision/text modality branch, $m$ are all from the vision/text modality. For the cross-modal modality branch, $X_{q,m}^l$ is from the text modality, while the other two are from the visual modality. Assume there are $L$ layers in total, we can stack all of the attention-based weight matrices in the multimodal pre-trained model, and derive the tensor $\mathcal{W}_0 = \{W_q^l, W_k^l, W_v^l\}_{l=1}^L \in \mathbb{R}^{d \times d \times N}$, where $d$ is the dimension of the embedding token, and $N$ denotes the total number of the weight matrices.

**Parameter Update.** For downstream tasks, directly updating the weight tensor with full fine-tuning will consume a huge amount of computation and brings about a heavy storage burden. We aim to update the knowledge with a few additional trainable parameters following the idea of PETL methods. Due to the redundancies of $\mathcal{W}_0$, we hope to implement the mode approximation of the tensor $\mathcal{W}_0$ to get the new learnable weight tensor $\Delta\mathcal{W}$ for downstream knowledge transfer. The

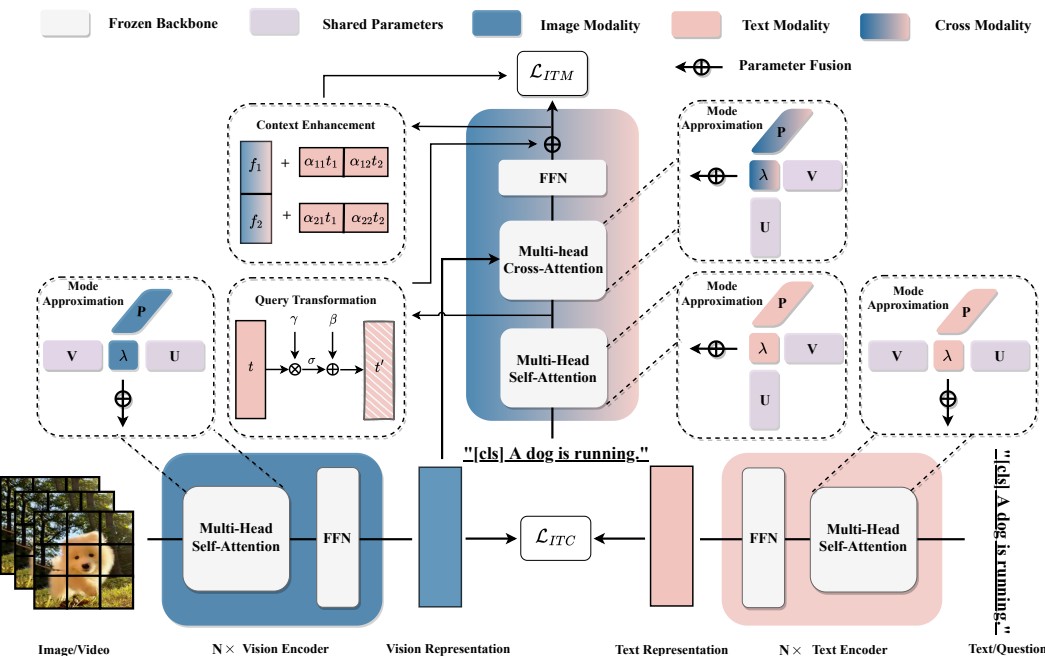

Figure 2: Demonstration of the overall framework. The frozen backbone network is shown in grey. The trainable parameters in color represent: blue for vision tasks, pink for text tasks, and the gradient color for fused modalities. Notably, globally shared parameters are represented in purple.

differences between our Aurora and existing PETL methods are demonstrated in Figure 1. Therefore, the backward propagation on the downstream training data $\mathcal{D}$ can be expressed as:

$$\nabla_{\mathcal{W}} = \nabla_{\Delta \mathcal{W}} = \frac{\partial \mathcal{L}\left(\mathcal{D}; \mathcal{W}_0 + \Delta \mathcal{W}\right)}{\partial \Delta \mathcal{W}} \tag{1}$$

### 3.2 Lightweight Design for PETL

To ensure a successful parameter-efficient tuning framework, it is important to prioritize a lightweight design that enables easy deployment and scalability. Recent research [45] has shown that when self-attention-based architecture is pre-trained on large-scale datasets, there is a significant amount of feature redundancy due to the limited responses of individual classes in a nearly infinite feature space. In fact, there exists a low-dimensional reparameterization that is as effective for fine-tuning as the full parameter space [2]. This inherent dimensionality describes the minimum dimension required to solve the optimization problem it defines to a certain level of precision. Considering the pre-trained parameters as a tensor, we recognize that approximation can effectively preserve low-rank yet discriminative non-redundant features, and scale down the weight tensor of pre-trained large-scale models along certain directions, thus making them more suitable for downstream tasks. Therefore, we propose a novel mode approximation method named Aurora, which utilizes mode approximation to update the frozen $\mathcal{W}_0$. Specifically, we borrow the idea of CANDECOMP/PARAFAC (CP) decomposition to decompose the learnable parameters $\Delta \mathcal{W}$ into a series rank-one tensors to explore the inherent dimensionality embedded in the features. The detailed architecture is shown in Figure 2.

**CP Decomposition.** In the typical tensor decomposition method, a 3D tensor has three modes (further introduced in Appendix), each of which can be viewed as the reduced projection of the tensor in a specific dimension. Given the 3D updated weight tensor $\Delta \mathcal{W} \in \mathbb{R}^{d \times d \times N}$, CP decomposition factorizes this tensor into a sum of $R$ rank-one components in total, and each component can be formalized as the outer product of three decomposed vectors in the formulation of:

$$\Delta \mathcal{W} = \sum_{r=1}^{R} \lambda_r (\boldsymbol{u_r} \circ \boldsymbol{v_r} \circ \boldsymbol{p_r}) \tag{2}$$

where $\boldsymbol{u_r} \in \mathbb{R}^d$, $\boldsymbol{v_r} \in \mathbb{R}^d$, and $\boldsymbol{p_r} \in \mathbb{R}^N$ are decomposed vectors for the $r$-th component, while each vector belongs to the corresponding mode matrix, i.e., $\boldsymbol{u_r}$ is the column vector in $U =$

$[\boldsymbol{u_1}, \cdots, \boldsymbol{u_r}, \cdots, \boldsymbol{u_R}]$. In addition, $\circ$ represents the outer product, $\lambda_r$ is the coefficient scalar of each component, and $R$ is the rank of CP decomposition. For better understanding, each component contributes to the value of the tensor in the sum of scalar products:

$$\boldsymbol{\Delta W}_{ijk} = \sum_{r=1}^{R} \lambda_r (u_{r,i} v_{r,j} p_{r,k}), \quad i,j \in \{1, 2, \cdots, d\}, k \in \{1, 2, \cdots, N\} \tag{3}$$

where $i, j, k$ denote the indices of the three modes.

**Mode Approximation.** In multimodal tasks, learning modality-specific representations and modality fusion representations are both important. Therefore, we aim to implement mode approximation to update the frozen attention-based weight tensor $\boldsymbol{\mathcal{W}}_0$, including the self-attention module in the vision/text encoders, and the cross-attention module in the multimodal encoders, which are based on the pre-trained multimodal foundation model like BLIP [41]. We first approximate the attention-based weight in these modules by initializing three mode factors, i.e., $U = [\boldsymbol{u_1}, \cdots, \boldsymbol{u_R}]$, $V = [\boldsymbol{v_1}, \cdots, \boldsymbol{v_R}]$, and $P = [\boldsymbol{p_1}, \cdots, \boldsymbol{p_R}]$. $U, P$ are randomly initialized with Gaussian distribution, and $V$ is with zeros, so that $\boldsymbol{\Delta W} = 0$ before training. It should be noted that $U, V$ are shared as global factors utilized for mode approximation, which means that our Aurora considers the cross-modal interaction and share the knowledge between these weight matrices in each modality. Besides, to further capture the discriminative features of each modality, we randomly initialize the learnable coefficient vector $\boldsymbol{\lambda}^m \in \mathbb{R}^R$ for each weight matrix on the modality $m$ respectively. With these three mode factors, we can implement the mode approximation in the forward propagation by the inverse progress of CP decomposition with input tensor $\boldsymbol{\mathcal{X}}^m$ as follows:

$$\boldsymbol{\mathcal{H}}^m = \boldsymbol{\mathcal{W}}_0 \boldsymbol{\mathcal{X}}^m + \left( \sum_{r=1}^{R} \lambda_r^m (\boldsymbol{u_r} \circ \boldsymbol{v_r} \circ \boldsymbol{p_r}) \right) \boldsymbol{\mathcal{X}}^m \tag{4}$$

Analyzed from the perspective of prompt learning, our idea of approximating the pre-trained weight parameters $\boldsymbol{\mathcal{W}}_0$ with additional trainable parameters $\boldsymbol{\Delta W}$ can be essentially understood as the soft prompts, which learns on downstream data based on CP decomposition. Such prompts not only provide better guidance for downstream tasks, but also are very lightweight in design, which greatly facilitates the application of pre-trained models to many cross-modal tasks in the unified mechanism.

### 3.3  Modality Alignment Design

Unlike existing methods that directly insert learnable networks to explicitly achieve cross-modal alignment, we further propose two effective modules to align different modalities with few trainable parameters. Thus, with the addition of mode approximation above, we can achieve a graceful prompt framework for cross-modal transfer, which is both lightweight and high-performance.

**Informative Context Enhancement.** For better modality alignment, we aim to provide prompts that can activate the fusion features after the cross-attention module. Inspired by the development in In-Context Learning [52, 18], we realize that the demonstration template is important for the prompts. The most intuitive approach is to align image-text pairs to obtain more cross-modal contextual information. However, even with relevant image regions, there may still be more than one way to describe these regions. Some texts may accurately summarize the content of an image, whereas others may not. Without *a priori* for matched textual information, we determine to introduce the context enhancement module to provide coverage of possible textual information.

We adopt the image-grounded text branch from BLIP and design a specific demonstration template for cross-modal prompt tuning. Given the fusion features of the image-grounded text branch $F = \{\boldsymbol{f_1}, \cdots, \boldsymbol{f_{|\mathcal{B}|}}\} \in \mathbb{R}^{|\mathcal{B}| \times E}$ and the textual query features of the self-attention module $T = \{\boldsymbol{t_1}, \cdots, \boldsymbol{t_{|\mathcal{B}|}}\} \in \mathbb{R}^{|\mathcal{B}| \times E}$, we utilize all of the query features with dimension $E$ in a batch $\mathcal{B}$ as the context for enhancement. Specifically, we calculate the attention score $\alpha \in \mathbb{R}^{|\mathcal{B}| \times |\mathcal{B}|}$ between feature $\boldsymbol{f_i}$ and each textual query feature from $\boldsymbol{t_1}$ to $\boldsymbol{t_{|\mathcal{B}|}}$ as:

$$\alpha_{ij} = \frac{\exp(\boldsymbol{f_i} \cdot \boldsymbol{t_j})}{\sum_{b=1}^{|\mathcal{B}|} \exp(\mathbf{f}_i \cdot \boldsymbol{t_b})} \tag{5}$$

In order to generate a more effective advanced fusion feature $F'$, all the adaptively weighted query features $T$ within a batch are collected together with a specific fusion feature $\boldsymbol{f_i}$ to form the demon-

Table 1: Results on image-text retrieval datasets MSCOCO and FLICKR30K. Using the text query, we simplify retrieving images as T→I and vice versa. Recall@$K$ represents the recall of top-$K$ returned samples. # Tunable is the size of the learnable parameters in the backbone network. $\Delta_{PETL}$ represents the performance gap between our Aurora and the best PETL method.

| Method | # Tunable | MSCOCO (I→T) | | | MSCOCO (T→I) | | | FLICKR30K (I→T) | | | FLICKR30K (T→I) | | |
|---|---|---|---|---|---|---|---|---|---|---|---|---|---|
| | | R@1 | R@5 | R@10 | R@1 | R@5 | R@10 | R@1 | R@5 | R@10 | R@1 | R@5 | R@10 |
| **Methods with full fine-tuning** | | | | | | | | | | | | | |
| UNITER | 330M | 65.7 | 88.6 | 93.8 | 52.9 | 79.9 | 88.0 | 87.3 | 98.0 | 99.2 | 75.6 | 94.1 | 96.8 |
| VILLA | 330M | - | - | - | - | - | - | 87.9 | 97.5 | 98.8 | 76.3 | 94.2 | 96.8 |
| OSCAR | 330M | 73.5 | 92.2 | 96.0 | 57.5 | 82.8 | 89.8 | - | - | - | - | - | - |
| ALIGN | 820M | 77.0 | 93.5 | 96.9 | 59.9 | 83.3 | 89.8 | 95.3 | 99.8 | **100.0** | 84.9 | 97.4 | 98.6 |
| ALBEF | 210M | 77.6 | 94.3 | 97.2 | 60.7 | 84.3 | 90.5 | 95.9 | 99.8 | **100.0** | 85.6 | 97.5 | **98.9** |
| BLIP | 223M | **81.9** | **95.4** | **97.8** | **64.3** | **85.7** | **91.5** | **97.3** | 99.9 | **100.0** | **87.3** | 97.6 | **98.9** |
| **Methods with frozen backbone** | | | | | | | | | | | | | |
| LoRA (r = 32) | 10.6M | 80.0 | 94.1 | 97.2 | 62.1 | 84.4 | 90.6 | 96.2 | 99.7 | 99.8 | 85.8 | 97.1 | 98.4 |
| UniAdapter (r=128) | 4.6M | 79.8 | 94.2 | 97.5 | 62.3 | 84.5 | 90.8 | 97.1 | **100.0** | **100.0** | 86.5 | 97.4 | 98.8 |
| UniAdapter (r=512) | 18.8M | 80.1 | 94.6 | 97.4 | 62.6 | 84.6 | 90.9 | 97.1 | 99.9 | **100.0** | 86.4 | 97.4 | **98.9** |
| Aurora (ours, r=64) | 0.1M | 80.2 | 95.1 | 97.7 | 62.4 | 84.5 | 91.0 | 96.8 | **100.0** | **100.0** | 86.7 | **97.8** | 98.7 |
| $\Delta_{PETL}$ | 0.5% | +0.1 | +0.5 | +0.2 | −0.2 | −0.1 | +0.1 | −0.3 | +0.0 | +0.0 | +0.2 | +0.4 | −0.2 |
| Aurora (ours, r=128) | 0.2M | 80.7 | 95.3 | **97.8** | 62.8 | 84.8 | 91.0 | 97.2 | **100.0** | **100.0** | 86.8 | 97.6 | **98.9** |
| $\Delta_{PETL}$ | 1% | +0.6 | +0.7 | +0.3 | +0.2 | +0.2 | +0.1 | +0.1 | +0.0 | +0.0 | +0.3 | +0.2 | +0.0 |

stration template. This form can adaptively absorb context query information to derive a better enhanced fusion feature for the image-text matching loss $\mathcal{L}_{\text{ITM}}$ as $\boldsymbol{f_i'} = \boldsymbol{f_i} + \sum_{j=1}^{|\mathcal{B}|} \alpha_{ij} \boldsymbol{t_j}$.

**Gated Query Transformation.** Another reason why modality alignment is difficult is that the multi-modality fusion branch network is deep, which can cause textual information to be lost during training. To address this issue, we propose a novel approach inspired by gating mechanisms to explicitly model the relative contributions of textual information during modality alignment. Specifically, instead of directly concatenating the fusion representation $\boldsymbol{f}$ (output of the Cross-Attention block) with the query representation $\boldsymbol{t}$ (output of the Self-Attention block) as residuals in existing methods [49], we learn a gated query function to balance the contributions of both modalities. Our gated query transformation involves two steps. First, we implement the transformation as follows: $\boldsymbol{t'} = \sigma(\gamma \odot \boldsymbol{t}) + \beta$, where $\gamma$ and $\beta$ are zero-initialized learnable transformation matrix and bias with an activation function $\sigma$. Second, we calculate the query gate $\mathbf{g}$ by computing the product of $\boldsymbol{t'}$ and $\boldsymbol{f}$ with Softmax. It should be noted that $\gamma$ and $\beta$ are zero initialized, so that $\boldsymbol{t'}$ is zero at the beginning of training. Hence, the query gate explicitly measures the contribution of the query representation in the formulation of $\mathbf{g} \odot \boldsymbol{f} + (\mathbf{1} - \mathbf{g}) \odot \boldsymbol{t'}$ to update the fusion representation $\boldsymbol{f}$.

## 4 Experiments

### 4.1 Experimental Settings

**Datasets & Baselines.** We evaluate Aurora on six benchmarks spanning four cross-modal tasks: image-text retrieval, question answering (QA), and video-text retrieval and QA. We compare Aurora with two types of approaches: full fine-tuning methods including SOTA for each task, and frozen backbone methods including LoRA [25] and UniAdapter [49]. See more details in Appendix.

**Implementation Details.** Our implementation is based on Salesforce's open-source codebase [41]. Following [49], we also apply BLIP-base [41] as our vision-language backbone for all the multimodal downstream tasks. We use PyTorch to implement all experiments on 8× NVIDIA V100 GPU (32G). We use AdamW [48] optimizer with a weight decay of 0.05 and a learning rate of 1e-4 obtained from grid search for all experiments. Note that during the fine-tuning process, the parameters of the backbone model are kept frozen. More training details can be seen in Appendix.

### 4.2 Performance Comparisons on Cross-modal Tasks

**Image-Text Retrieval.** Table 1 shows performances for image-text retrieval tasks on MSCOCO [47] and FLICKR30K [54]. It can be observed that our Aurora (R=64) achieves comparable results with the state-of-the-art frozen backbone method while using only 0.5% of its parameters. When we

Table 2: Results on two benchmark video-text retrieval datasets, MSR-VTT and DiDemo. Recall@$K$ and MdR are utilized as the evaluation metric, where MdR measures the median rank of target items in the retrieved ranking list. Input means the sampling number and frame shape of each video.

| Method | Input | # Pretrain | # Tunable | MSR-VTT | | | | DiDemo | | | |
| --- | --- | --- | --- | --- | --- | --- | --- | --- | --- | --- | --- |
| | | | | R@1 | R@5 | R@10 | MdR | R@1 | R@5 | R@10 | MdR |
| **Methods with full fine-tuning** | | | | | | | | | | | |
| ClipBERT | 16×448 | 5M | 135M | 22.0 | 46.8 | 59.9 | 6.0 | 20.4 | 48.0 | 60.8 | 6.0 |
| Frozen in Time | 32×224 | 5M | 180M | 31.0 | 59.5 | 70.5 | 3.0 | 34.6 | 65.0 | 74.7 | 3.0 |
| ALPRO | 8×224 | 5M | 245M | 33.9 | 60.7 | 73.2 | 3.0 | 35.9 | 67.5 | 78.8 | 3.0 |
| VIOLET | 5×224 | 138M | 306M | 34.5 | 63.0 | 73.4 | - | 32.6 | 62.8 | 74.7 | - |
| All-in-one | 9×224 | 138M | 110M | 37.9 | 68.1 | 77.1 | - | 32.7 | 61.4 | 73.5 | 3.0 |
| CLIP4Clip | 12×224 | 400M | 124M | 43.1 | 70.4 | 80.8 | 2.0 | 42.8 | 68.5 | 79.2 | 2.0 |
| CLIP-Hhiker | 120×224 | 400M | 124M | 47.7 | **74.1** | **82.9** | - | - | - | - | - |
| **Methods with frozen backbone** | | | | | | | | | | | |
| CLIP-Prompt | 16×224 | 400M | 64M | 36.7 | 64.6 | - | - | - | - | - | - |
| LoRA (r=32) | 8×224 | 129M | 10.6M | 49.9 | 72.0 | 81.3 | 2.0 | 50.9 | 75.3 | 82.4 | 2.0 |
| UniAdapter (r=128) | 8×224 | 129M | 4.6M | 49.7 | 71.9 | 81.5 | 2.0 | 49.0 | 75.5 | 83.3 | 2.0 |
| UniAdapter (r=512) | 8×224 | 129M | 18.8M | 50.6 | 73.4 | 81.6 | **1.0** | 52.1 | 77.3 | 85.2 | **1.0** |
| Aurora (ours, r=64) | 8×224 | 129M | **0.1M** | **52.4** | 73.9 | 82.0 | **1.0** | **53.1** | **77.4** | **85.3** | **1.0** |
| $\Delta_{PETL}$ | - | - | 0.5% | +1.8 | +0.5 | +0.4 | +0.0 | +1.0 | +0.1 | +0.1 | +0.0 |

Table 3: Results on two visual question answering datasets, VQAv2 and MSRVTT-QA. # Tunable represents the number of learnable parameters. For VQAv2, we report the test-dev and test-std results, for MSRVTT-QA, accuracy is used as the evaluation metric.

| Method | # Tunable | VQAv2 | | Method | # Tunable | MSRVTT-QA |
| --- | --- | --- | --- | --- | --- | --- |
| | | test-dev | test-std | | | test acc |
| **Methods with full fine-tuning** | | | | | | |
| VL-T5/BART | 165M | - | 71.30 | CLIPBERT | 135M | 37.4 |
| SOHO | 155M | 73.25 | 73.47 | ALPRO | 245M | 42.1 |
| OSCAR | 330M | 73.61 | 73.82 | Just-Ask | 200M | 41.5 |
| UNITER | 330M | 73.82 | 74.03 | VIOLET | 306M | 43.9 |
| ALBEF | 266M | 75.84 | 76.04 | MERLOT | 233M | 43.1 |
| BLIP | 337M | 77.44 | 77.48 | All-in-one | 110M | 44.3 |
| **Methods with frozen backbone** | | | | | | |
| UniAdapter (r=128) | 4.6M | 73.72 | 73.71 | UniAdapter (r=128) | 4.6M | 44.2 |
| UniAdapter (r=512) | 18.8M | 75.44 | 75.56 | UniAdapter (r=512) | 18.8M | 44.7 |
| Aurora (ours, r=64) | **0.1M** | **77.69** | **77.87** | Aurora (ours, r=64) | **0.1M** | **44.8** |
| $\Delta_{PETL}$ | - | +2.25 | +2.31 | $\Delta_{PETL}$ | - | +0.1 |

increase the rank to 128, Aurora can further boost the performance, surpassing all the frozen backbone methods, and even outperforming some full fine-tuning methods with fewer trainable parameters.

**Video-Text Retrieval.** To further verify the performance of our Aurora in the field of video-text retrieval, we conduct experiments on two video datasets, MSRVTT [69] and DiDemo [5], and the results are presented in Table 2. With only about 0.1M trainable parameters, our Aurora directly achieves better performances than all the frozen backbone methods, and we outperform most full fine-tuning methods. This indicates that our Aurora has an excellent understanding ability under video-text scenes, even with relatively few trainable parameters.

**Visual Question Answering.** In an effort to further explore the potential of our Aurora, we evaluate it on VQA/VideoQA tasks with VQAv2 [24]/MSRVTT-QA [68] datasets and demonstrate the evaluation results in Table 3. Unlike retrieval tasks, VQA task needs to verify the model's multimodal generative ability. We share the trainable parameters of the multimodal encoder and multimodal decoder to further reduce the parameter amount. From the results, we find that our Aurora outperforms UniAdapter and all the full fine-tuning methods, indicating that Aurora can have powerful transfer ability for downstream generative tasks.

## 4.3 Performance Comparisons on Zero-shot Setting

To evaluate the generalization ability of Aurora, we conduct experiments on cross-modal tasks with zero-shot setting, and make comparisons with the pretrained version, full fine-tuning method, LoRA, and UniAdapter.

Table 4: Zero-shot performance analysis.

| Method | # Parameter | MSRVTT (T→V) | | | DiDemo (T→V) | | |
|---|---|---|---|---|---|---|---|
| | | R@1 | R@5 | R@10 | R@1 | R@5 | R@10 |
| BLIP | 223M | 41.5 | 62.0 | 70.7 | 42.1 | 59.6 | 67.3 |
| BLIP + FFT | 223M | 42.5 | 62.8 | 71.6 | 43.0 | 60.5 | 68.3 |
| BLIP + LoRA | 10.6M | 42.7 | 62.8 | 71.4 | 43.4 | 60.3 | 68.2 |
| BLIP + UniAdapter | 18.8M | 42.2 | 62.6 | 71.1 | 43.1 | 60.2 | 67.9 |
| BLIP + Aurora | **0.1M** | **43.1** | **63.5** | **72.0** | **44.6** | **61.4** | **68.6** |

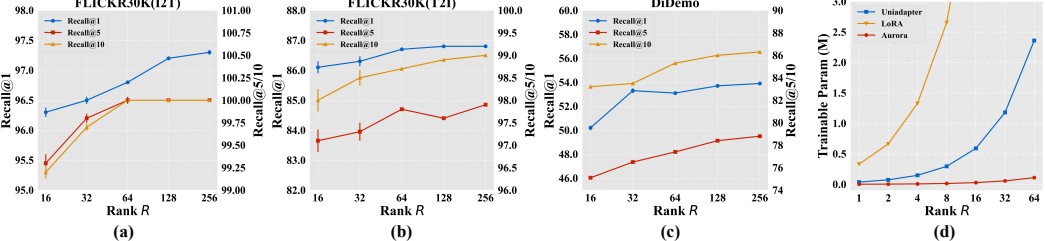

Figure 3: The answer to how rank $R$ affects Aurora. **(a)**, **(b)**, and **(c)** show the performance increase accompanied with larger $R$ on three different cross-modal tasks. Notably, our results are divided on two $y$-axes for clear demonstration, where Recall@1 is shown on the left axis and Recall@5/10 are on the right one. **(d)** compares the parameter scalability with other PETL methods.

Specifically, experiments are based on the pre-trained large-scale multimodal foundation models, i.e., BLIP, which is further tuned in each corresponding method on MSCOCO. Then, we utilize two video datasets as the zero-shot data for validation. Table 4 provides an overview of the performance of Aurora on various zero-shot multimodal tasks. It is obvious that Aurora achieves the highest zero-shot performance while requiring the least number of trainable parameters during vision-language pre-training, which represents more powerful general-purpose understanding ability.

## 4.4 Analysis of Different Designs

We thoroughly evaluate how the core operations in Aurora affect results, i.e., the rank $R$, parameter sharing, context enhancement, and the gated query transformation. We conduct experiments to analyze the impacts of different designs, which are implemented on several cross-modal tasks.

**How Rank of CP Decomposition Affects Aurora?**

Results on Flickr30K and DiDemo are shown in the left three columns in Figure 3. A fundamental conclusion is that as $R$ increases, the dimensionality of the model representation also increases, leading to better performance on downstream tasks. However, when $R$ reaches a certain range, the rate of increase slows down, which may be related to the redundancy of high-dimensional features.

In addition, we show in (d) of Figure 3 that as $R$ increases, the growth rate of parameter size in Aurora is much slower than that of LoRA and UniAdapter. As another tensor decomposition method, our scalability and ease of deployment are much stronger than the baselines.

**How Does Aurora Benefit from Informative Context Enhancement?**

For an in-depth investigation, we introduce three ablated variants, i.e., **Aurora w/o C** removes the context enhancement, **Aurora w. R** replaces the context with random vectors, and **Aurora w. M** simply takes the average features of all queries. Ablation results are shown in Figure 4, where we separate the results on two $y$-axes respectively, which can better demonstrate the differences across different indicators more clearly. And the results of our Aurora are marked by the black upper lines. Surprisingly, using random vectors to supplement fused features still achieves better

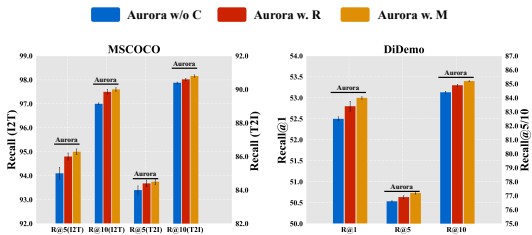

Figure 4: Analysis of the impact of the informative context enhancement module.

results than removing this module. Moreover, we observe that adaptive weight $\alpha$ causes better performance than the uniform weight. These phenomena fully demonstrate that with few parameters,

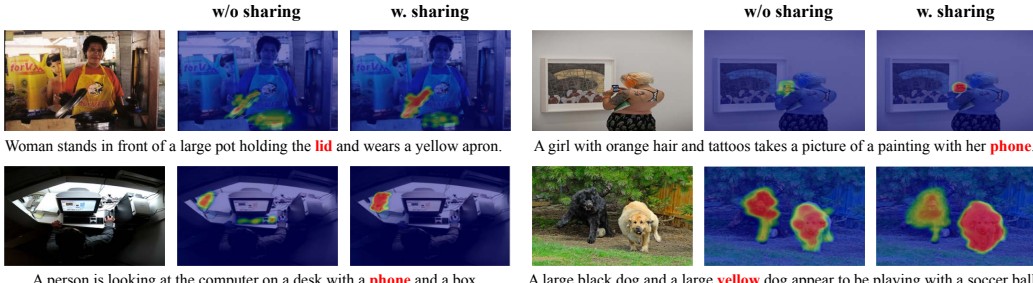

Figure 5: Visualization of cross-attention map comparisons on Flickr30K, which shows the capability to locate the most semantic-related visual parts for specific words in the text.

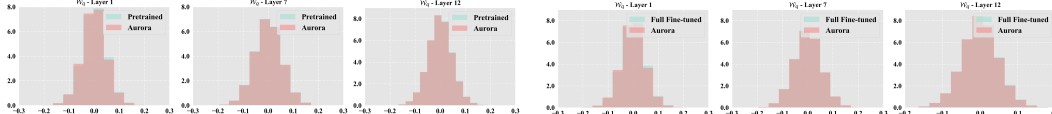

Figure 7: The left three columns show the parameter distribution of the pre-trained large-scale multimodal foundation model (BLIP) *vs.* our Aurora, which is tuned on MSCOCO. And the right part is full fine-tuned model *vs.* Aurora. Notably, $\mathcal{W}_q$ is the stack of the query projection matrices in different modality branches.

supplementing more context information to fused features is crucial to promote modality alignment for downstream tasks, and our context enhancement is the optimal choice.

**How Does Aurora Benefit from Parameter Sharing?**

See Figure 5 for the effectiveness of parameter sharing. We observe that the parameter sharing strategy can achieve a better capability to locate the most semantic-related visual parts for specific words in the text. It should be noted that we are able to reduce 0.4M parameters through parameter sharing. Moreover, we also conduct quantitative experiments on FLICKR30K, in which parameter sharing results in better performance with an average improvement of 0.1. Therefore, we can draw the conclusion that parameter sharing during training not only decreases the parameter dependence but also boost the cross-modal interaction for better performances.

**How Does Aurora Benefit from Gated Query Transformation?**

To further validate the effectiveness of the Gated Query Transformation module, we conduct a comprehensive ablation study on MSCOCO and DiDemo datasets with two variants. The experimental results are shown in Figure 6, where **Aurora w/o Q** (blue bars) represents the removal of the Gated Query Transformation and **Aurora w. O** (red bars) initializes the transformation matrix and bias with ones. The gap between the blue bars and ours shows that this module helps promote the final performance to some extent, which also indicates that adaptively incorporat-

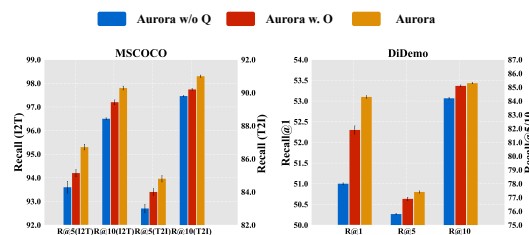

Figure 6: Analysis of the impact of the gated query transformation module.

ing text information into the modality fusion branch has a positive effect on modality alignment under the deep network architecture. In addition, the difference between the red bars and ours shows that initializing from zeros is more conducive to adaptively controlling the proportion of text feature fusion, and can gradually achieve better modality alignment for better results.

### 4.5 Visualization Analysis

**Parameter Distribution.** We visualize the parameter distributions of the pre-trained model, fully fine-tuned model, and our Aurora in Figure 7. Aurora involves gradually embedding the knowledge learned from downstream tasks into the parameters of a pre-trained model, while amplifying the original parameters in a certain direction. From the left part, we can see that our Aurora only adjusts the pre-trained model parameters in a small local range, but it can have a better effect on downstream

tasks. Meanwhile, we can see from the right part that Aurora's parameter distribution is very close to that of the fully fine-tuned model, with only small changes in a small range. These small changes enable our method to achieve superior performance on many downstream tasks.

## 5  Conclusion

This paper proposes Aurora, a graceful prompt framework for cross-modal transfer. We first leverage mode approximation to implement multimodal prompt tuning, which explores the low intrinsic dimension with only 0.04% parameters of the pretrained model. Then, to better reduce the modality gap, we propose Informative Context Enhancement and Gated Query Transformation modules under extremely low parameter scenarios. Extensive evaluation of Aurora on six cross-modal benchmarks shows that it not only outperforms the state-of-the-art but even surpasses full fine-tuning approach.

**Limitations.** The representation ability of our Aurora to some extent depends on the setting of the rank $R$. However, it also unavoidably increases the parameter size and thus the computational cost. Therefore, it is crucial for us to select a proper rank in the trade-off between parameter size and performance. Unfortunately, it is challenging for us to determine a suitable rank beforehand with the variations of downstream tasks and data sizes, which poses a challenge to the application of our model. We will further optimize this issue in future work.

**Broader Impact.** Multimodal pre-trained large models have wide applications in the real world, including image-text understanding, retrieval, question answering, and generation. As models scale up, they will achieve more amazing performances. Our Aurora enables lightweight transfer of large-scale models, which allows us to better utilize cloud-based large models and guide them to produce high-performance edge applications under constrained computational resources.

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

# APPENDIX

To provide a comprehensive demonstration of our approach, we will supplement additional details. The arrangement of these sections is as follows: First, we demonstrate the core concepts of our Aurora for clarity. Second, we comprehensively make overall comparisons with existing methods on various multimodal tasks. Then, we provide details regarding the datasets and baselines in Section C, while the concrete training details are outlined in Section D. We then conduct a comprehensive analysis of the computational costs, including time and memory consumption, along with algorithmic complexity in Section E. Furthermore, we provide theoretical support for our approach in Section F. Finally, we present visualizations of our proposed Aurora for several cross-modal tasks to facilitate qualitative comparisons in Section G. To represent our method clearly and concisely, we use lowercase letters for scalars, bold lowercase letters for **vectors**, italicized uppercase letters for *MATRICES*, and bold italicized uppercase letters for ***TENSORS*** in the equations, respectively.

## A   Concept Definition

According to [1, 51], we will offer a precise definition of the fundamental notions underpinning our key Mode Approximation component.

First, the definition of tensors can be demonstrated as follows:

**Definition 1** *(Tensor). Let $\mathcal{D}_1, \mathcal{D}_2, \cdots, \mathcal{D}_N \in \mathbb{N}$ denote index upper bounds, a tensor $\boldsymbol{\mathcal{W}} \in \mathbb{R}^{\mathcal{D}_1 \times \cdots \times \mathcal{D}_N}$ of order $N$ is an $N$-way array where elements $\mathcal{W}_{d_1, d_2, \cdots, d_n}$ are indexed by $d_n \in \{1, 2, \cdots, \mathcal{D}_n\}, for\ 1 \leq n \leq N.$*

Then, the concept of the mode is formulated as follows:

**Definition 2** *(Mode). Let $\boldsymbol{\mathcal{W}} \in \mathbb{R}^{n_1 \times n_2 \times \cdots \times n_d}$ be a d-dimensional tensor. The mode-k matricization of $\boldsymbol{\mathcal{W}}$, denoted as $\mathcal{W}^{(k)} \in \mathbb{R}^{n_k \times (n_1 \cdots n_{k-1} n_{k+1} \cdots n_d)}$, is obtained by unfolding the tensor along its k-th mode and arranging the entries as rows in a matrix.*

Given the mathematical definition of the mode, we can implement decomposition in the context of CP decomposition. We stack all the weight matrices in the attention layer (i.e., $W_q, W_k, W_v$) of all the branches into a tensor, which needs to be updated as $\Delta \boldsymbol{\mathcal{W}}$. Since our method assumes that the stack of weight matrices is a three-order tensor, $k$ is three in our settings, and thus the CP decomposition can be illustrated as follows:

$$\Delta \boldsymbol{\mathcal{W}} = \sum_{r=1}^{R} \lambda_r \mathbf{u}_r \circ \mathbf{v}_r \circ \mathbf{p}_r, \tag{6}$$

where $R$ is the rank, $\lambda_r$ are non-negative scalar weights, and $\mathbf{u}_r \in \mathbb{R}^{n_1}$, $\mathbf{v}_r \in \mathbb{R}^{n_2}$, $\mathbf{p}_r \in \mathbb{R}^{n_3}$ are non-zero factor vectors. And the mode-$k$ unfolding of the tensor $\Delta \boldsymbol{\mathcal{W}}$ is $U, V, P$ respectively.

Our proposed Aurora aims to approximate the latent mode matrix with randomly initialized learnable parameters, which can learn knowledge on downstream tasks in a lightweight manner.

## B   Overall Comparison

We compare all the baselines for three downstream tasks and presented a comprehensive illustration in Figure 8. The arrow in the figure points towards better performance on dual metrics, as it moves towards the upper right corner. We rank the size of the model parameters and use it as a basis for determining the size of the bubbles, which are also displayed in Figure 8. It is evident that our method performs remarkably well even with smaller parameter sizes, and in several instances, outperforms the fully fine-tuned approach, demonstrating the strength of our architecture.

## C   More Details for the Baselines & Datasets.

**Baselines.** For Frozen Backbone methods, UniAdapter [49] is currently the state-of-the-art method for parameter-efficient transfer learning in the field of multimodality and can be considered as a

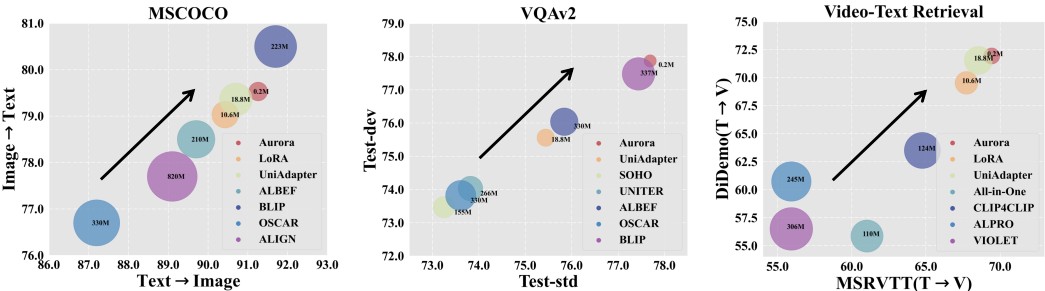

Figure 8: Performance-Parameter comparison of different methods on each multimodal downstream task. Note that the bubble size denotes the size of total tunable parameters.

representative of the Adapter class of methods in the multimodal domain. LoRA [25] is another important branch of parameter-efficient transfer learning methods. Its core idea is to use low-rank decomposed matrices to calculate the incremental change of model parameters during adaptation on downstream tasks. To enable comparison with a wider range of baselines, we replicate many of the settings from prior works and reuse their experiment results whenever possible. It should be noted that this means some baselines only appear in specific experiments.

As for Full Fine Tuning methods, we apply UNITER [13], VILLA [23], OSCAR [43], ALIGN [29], ALBEF [42] and BLIP [41] for image-text retrieval tasks, then we use ClipBERT [37], Frozen in Time [6], ALPRO [40], VIOLET [22], All-in-one [65], CLIP4Clip [50] and CLIP-Hhiker [7] for text-video retrieval tasks, finally we adopt ClipBERT[37], ALPRO [40], Just-Ask [70], VIOLET [22], MERLOT [75], All-in-one [65] for VideoQA task while adopt VL-T5/BART [14], SOHO [27], OSCAR [43], UNITER [13], ALBEF [42] and BLIP [41] for VQA tasks.

**Datasets.** We provide a comprehensive introduction to the datasets of various downstream tasks in the multimodal scenario, as outlined below:

- **MSCOCO** [47] is a large scale image-text dataset and each image is annotated with five captions. Following [49, 34], we use Karpathy split of MSCOCO: 5,000 images for testing, 5,000 images for validation, and the rest for training.

- **Flickr30K** [54] contains 31,000 images collected from Flickr. Each image is usually referenced with five human annotations. Following previous works [49, 21], we use 1,000 images for testing, another 1000 for validation, and the rest for training.

- **MSR-VTT** [69] contains 10,000 video clips and each video clip is annotated with 20 English sentences. Following recent works [49, 50], we adopt 1k-test split for training and testing.

- **DiDemo** [5] is one of the most commonly used datasets for the temporal localization of events in videos. It contains about 10,000 videos and 40,000 annotations. we follow [49, 6] to concatenate all descriptions corresponding to the same video into a single sentence to conduct actually paragraph-to-video retrieval task.

- **VQAv2** [24] is one of the most famous visual question answering datasets which contains 83k/41k/81k images for training/validation/testing. Following [49, 42, 41], we use both training and validation splits of VQAv2 and additional training samples from Visual Genome [36] for training. The results should be evaluated by the official server and we report the results on the test-dev and test-std splits.

- **MSRVTT-QA** [68] is one of the most popular video question answering datasets. It's constructed based on MSRVTT and has 243k open-ended questions associated with 10k videos. We follow [49, 37] to employ the standard split for training and testing.

## D  More Training Details

### D.1  Frozen Backbone

Another important design in BLIP is CapFlit, It contains a Captioner to generate captions given web-searched images and a Filter to remove noisy image-text pairs, both Captioner and Filter are finetuned

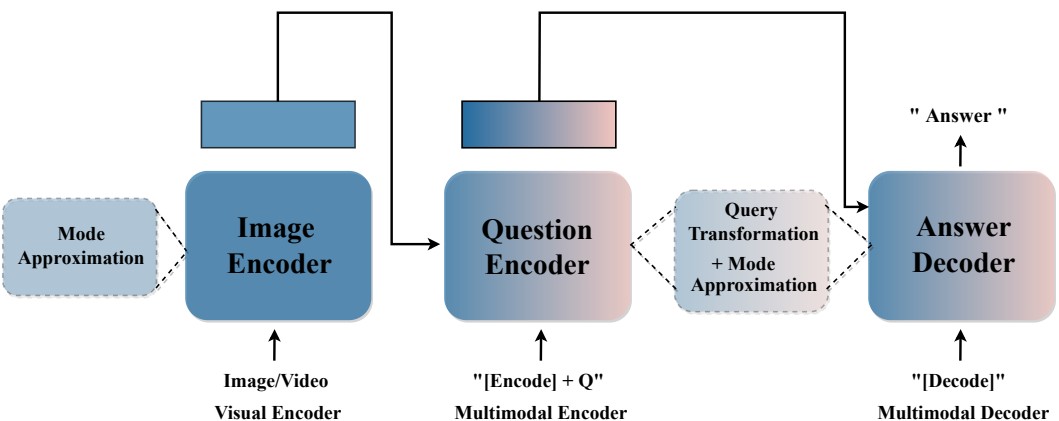

Figure 9: Model architecture for the Visual Question Answering tasks on both images and videos.

individually on the COCO dataset while using different objective loss. In addition, BLIP uses momentum technology to further improve the correctness of the image-text matching relationship.

### D.2 Architecture for VQA Tasks

Figure 9 shows the architecture of Aurora for Visual Question Answering tasks. Compared to Retrieval tasks, the VQA architecture has an additional answer decoder. During fine-tuning, the images/videos are first encoded by a single-modal visual encoder, and then the image/video-text pairs are fused using a multimodal encoder and given to the decoder for prediction. Answers are used as ground truth and Language-Model Loss is utilized for parameter updating throughout the entire training process. As the ITM Loss is no longer needed, we remove the Informative Context Enhancement module from the VQA architecture. Meanwhile, we retain the Gated Query Transformation module to preserve the complete semantic information of the question representations as much as possible. Finally, in order to further reduce the number of parameters, we share the learnable parameters of the multimodal encoder and the multimodal decoder.

### D.3 Implementation Details

In this section, we give more training details about our Aurora.

- For image-text downstream tasks, we set the image size into 384×384. We use cosine decay to update the learning rate during training. We set the batch size to 16 for each GPU and train a total of 6 epochs.

- For video-text retrieval tasks, we randomly sample T = 8(16) frames for each video during training(testing) while setting the frame size to be 224×224. We use cosine decay to update the learning rate, we set the batch size to 8 per GPU during training and train for a total of 5 epochs.

- For VideoQA task, we also sample 8 frames per video for training while the frame size is changed to 384×384. While training, we set the batch size to 4 per GPU and train 10 epochs in total. During the evaluation, we randomly sample 16 frames for each video, and we use greedy search to generate the next token when producing answer for its corresponding question.

- For VQA task, we set the image size to 480×480 for training/inference and adopt a batch size of 16 for each GPU. We also adopt cosine decay to change the value of learning rate at different epochs and we train 10 epochs.

We also perform a simple cleaning of the text data and truncate all words beyond the maximum length of the sentence. All data processing and partitioning are consistent with UniAdapter and LoRA to ensure fair comparison. When implementing CP decomposition, textual encoder, visual encoder, and multimodal encoder share the same global mode factor $U$ and factor $V$ to do parameter sharing and we initialize the weights of factor $V$ to be zero.

Table 5: Training time and GPU memory comparison.

| Method | #Tunable | MSCOCO | | FLICKR30K | | MSRVTT-QA | | VQAv2 | | DiDemo | | MSRVTT | |
|---|---|---|---|---|---|---|---|---|---|---|---|---|---|
| | | Time | Memory | Time | Memory | Time | Memory | Time | Memory | Time | Memory | Time | Memory |
| UniAdapter (r=512) | 18.8M | 1.00 | 1.00 | 1.00 | 1.00 | 1.00 | 1.00 | 1.00 | 1.00 | 1.00 | 1.00 | 1.00 | 1.00 |
| UniAdapter (r=128) | 4.6M | 0.86 | 0.95 | 0.93 | 0.95 | 0.89 | 0.91 | 0.79 | 0.92 | **0.88** | 0.94 | **0.94** | 0.94 |
| Aurora (r=128) | 0.2M | 0.90 | **0.93** | 0.93 | **0.94** | 0.88 | **0.89** | 0.79 | 0.91 | 0.90 | 0.92 | 0.95 | 0.93 |
| Aurora (r=64) | 0.1M | **0.84** | **0.93** | **0.92** | **0.94** | **0.86** | **0.89** | **0.76** | **0.89** | **0.88** | **0.88** | **0.94** | **0.92** |

# E   Cost Analysis

In Table 5, following [49], we report the training time and GPU memory cost for both retrieval and VQA tasks. We regard the training time and memory cost of UniAdapter(r=512) as one unit. Since we adopt the same backbone models, the forward propagation process of the two methods, Aurora and UniAdapter, is almost consistent, and the time cost is similar. However, our Aurora has fewer trainable parameters, resulting in a slightly smaller GPU memory footprint.

Then, We will give a theoretical analysis of the parametric complexity of the three PETL methods. Assume that the frozen backbone's visual, textual, and multimodal encoder all have L transformer layers while multimodal encoders contain cross-attention modules and visual, textual encoders contain self-attention modules. We only approximate the Query/Key/Value weight matrix in these attention-based modules. Let d donates the dimension of the hidden feature and r for rank, so the parametric complexity of the LoRA is $L \times (3 + 6) \times 2dr \sim \mathcal{O}(Ldr)$, the complexity of the UniAdapter is $L \times 4dr \sim \mathcal{O}(Ldr)$, and the complexity of our Aurora is $L \times (3 + 6) \times 2r + 2dr + 2Ld \sim \mathcal{O}((L+d)r)$, normally r and L are much smaller than d, so from the above analysis we can draw the conclusion that when r increases, our Aurora can achieve the lowest parameter cost.

# F   Theoretical Analysis

Write $\Delta \mathcal{W} = \sum_{r=1}^{R} \lambda_r (\boldsymbol{u_r} \circ \boldsymbol{v_r} \circ \boldsymbol{p_r})$, and the $(i, j, k)$-element of $\Delta \mathcal{W}$ is:

$$\Delta \mathcal{W}_{ijk} = \sum_{r=1}^{R} \lambda_r u_{ri} v_{rj} p_{rk}. \tag{7}$$

Recall that the *Frobenius norm* of a tensor $\mathcal{X} \in \mathbb{R}^{d \times d \times N}$ is given by:

$$\|\mathcal{X}\|_F = \left( \sum_{i_1=1}^{d} \sum_{i_2=1}^{d} \sum_{i_3=1}^{N} |x_{i_1,i_2,i_3}|^2 \right)^{1/2}, \tag{8}$$

Hence, it suffices to analyze the convergence rate of our parameter tensors in the Euclidean space $\mathbb{R}^{ddN}$, where $ddN$ denotes the product of $d \times d \times N$ in order to distinguishing from the space of multi-dimensional arrays of size $d \times d \times N$.

**Assumption F.1** *We identify $\mathbb{R}^{d \times d \times N}$ with $\mathbb{R}^{ddN}$, and let the loss function $\mathcal{L}$ be defined on $\mathbb{R}^{ddN}$, while we still write $\mathcal{L}(\mathcal{W})$ where $\mathcal{W} \in \mathbb{R}^{d \times d \times N}$ is a tensor. We will also use Frobenius norm and $\ell^2$ norm interchangeably, which means that:*

$$\|\mathcal{X}\|_F = \|\mathcal{X}\|_2. \tag{9}$$

**Assumption F.2** *We assume that the loss function $\mathcal{L} : \mathbb{R}^{ddN} \to \mathbb{R}$ has the following property:*

1. *$\mathcal{L}$ is injective.*

2. *$\mathcal{L}$ is strongly convex: there exist $m$ and $M$ such that:*

$$mI \preceq \nabla^2 \mathcal{L}(\mathcal{X}) \preceq MI. \tag{10}$$

*That is, $\nabla^2 \mathcal{L}(\mathcal{X}) - mI$ is positive semidefinite and $\nabla^2 \mathcal{L}(\mathcal{X}) - MI$ is negative semidefinite.*

Let $\{\lambda_r^{(0)}, \boldsymbol{u}_r^{(0)}, \boldsymbol{v}_r^{(0)}, \boldsymbol{p}_r^{(0)} : r = 1, \cdots, R\}$ be the randomly initialized vectors used for tensor decomposition, where $\lambda_r^{(0)} \in \mathbb{R}, \boldsymbol{u}_r^{(0)} \in \mathbb{R}^d, \boldsymbol{v}_r^{(0)} \in \mathbb{R}^d, \boldsymbol{p}_r^{(0)} \in \mathbb{R}^N$. Denote

$$\Delta \boldsymbol{\mathcal{W}}^{(0)} = \boldsymbol{\mathcal{W}}_0 = \sum_{r=1}^{R} \lambda_r^{(0)} \boldsymbol{u}_r^{(0)} \circ \boldsymbol{v}_r^{(0)} \circ \boldsymbol{p}_r^{(0)}. \tag{11}$$

Let $\Delta \boldsymbol{\mathcal{W}}^{(n)}$ be the parameter tensor returned by the $n$th training epoch, that is,

$$\Delta \boldsymbol{\mathcal{W}}^{(n)} = \sum_{r=1}^{R} \lambda_r^{(n)} \boldsymbol{u}_r^{(n)} \circ \boldsymbol{v}_r^{(n)} \circ \boldsymbol{p}_r^{(n)}. \tag{12}$$

**Theorem F.1** *Under the above assumptions, and suppose that we train for $n$ epochs with $\eta \leq 1/M$ using gradient descent. Let $\boldsymbol{\mathcal{W}}^*$ be the optimal parameter tensor, then,*

$$\mathcal{L}\left(\boldsymbol{\mathcal{W}}_0 + \Delta \boldsymbol{\mathcal{W}}^{(n)}\right) \to \mathcal{L}\left(\boldsymbol{\mathcal{W}}^*\right) \quad (n \to \infty). \tag{13}$$

*Moreover, $\boldsymbol{\mathcal{W}}^*$ is unique.*

**Proof F.1** *The proof follows from [8]. For notational convenience let $\mathcal{X}^{(n)} = \boldsymbol{\mathcal{W}}_0 + \Delta \boldsymbol{\mathcal{W}}^{(n)}$, and denote the optimal value $\mathcal{L}(\boldsymbol{\mathcal{W}}^*)$ by $\lambda^*$. We will begin by analyzing the convergence using arbitrary $\mathcal{X}, \mathcal{Y} \in \mathbb{R}^{ddN}$, and then plug in our parameter tensors. By Taylor's theorem we can write:*

$$\mathcal{L}(\mathcal{Y}) = \mathcal{L}(\mathcal{X}) + \nabla \mathcal{L}(\mathcal{X})^T (\mathcal{Y} - \mathcal{X}) + \frac{1}{2}(\mathcal{Y} - \mathcal{X})^T \nabla^2 \mathcal{L}(\mathcal{Z})(\mathcal{Y} - \mathcal{X}), \tag{14}$$

*where $\mathcal{Z}$ lies in the line segment joining $\mathcal{X}$ and $\mathcal{Y}$. By the strong convexity assumption, we have,*

$$\frac{1}{2}(\mathcal{Y} - \mathcal{X})^T \nabla^2 \mathcal{L}(\mathcal{Z})(\mathcal{Y} - \mathcal{X}) \geq \frac{1}{2}(\mathcal{Y} - \mathcal{X})^T m(\mathcal{Y} - \mathcal{X}) = \frac{m}{2} \|\mathcal{Y} - \mathcal{X}\|_2^2. \tag{15}$$

*Hence*

$$\mathcal{L}(\mathcal{Y}) \geq \mathcal{L}(\mathcal{X}) + \nabla \mathcal{L}(\mathcal{X})^T (\mathcal{Y} - \mathcal{X}) + \frac{m}{2} \|\mathcal{Y} - \mathcal{X}\|_2^2. \tag{16}$$

*Now we use $\|\nabla \mathcal{L}(\mathcal{X})\|_2$ to bound $\mathcal{L}(\mathcal{X}) - \lambda^*$. The right-hand side of (16) is a convex quadratic function of $\mathcal{Y}$, hence $\mathcal{Y}^* = \mathcal{X} - 1/m \nabla \mathcal{L}(\mathcal{X})$ is the minimizer, thus,*

$$\mathcal{L}(\mathcal{Y}) \geq \mathcal{L}(\mathcal{X}) + \nabla \mathcal{L}(\mathcal{X})^T (\mathcal{Y}^* - \mathcal{X}) + \frac{m}{2} \|Y^* - \mathcal{X}\|_2^2 \tag{17}$$

$$= \mathcal{L}(\mathcal{X}) + \nabla \mathcal{L}(\mathcal{X})^T \left(-\frac{1}{m} \nabla \mathcal{L}(\mathcal{X})\right) + \frac{m}{2} \left\|\frac{1}{m} \nabla \mathcal{L}(X)\right\|_2^2 \tag{18}$$

$$= \mathcal{L}(\mathcal{X}) - \frac{1}{2m} \|\nabla \mathcal{L}(\mathcal{X})\|_2^2. \tag{19}$$

*Since $\mathcal{Y}$ is arbitrary, plugging $\mathcal{X} = \mathcal{X}^{(n)}$, we have*

$$\lambda^* \geq \mathcal{L}(\mathcal{X}^{(n)}) - \frac{1}{2m} \left\|\nabla \mathcal{L}(\mathcal{X}^{(n)})\right\|_2^2 \tag{20}$$

*By the assumption $\nabla^2 \mathcal{L}(\mathcal{X}) \preceq MI$ we have*

$$\mathcal{L}(\mathcal{Y}) \leq \mathcal{L}(\mathcal{X}) + \nabla \mathcal{L}(\mathcal{X})^T (\mathcal{Y} - \mathcal{X}) + \frac{M}{2} \|\mathcal{Y} - \mathcal{X}\|_2^2. \tag{21}$$

*Plugging in $\mathcal{Y} = \mathcal{X}^{(n)} - \eta \nabla \mathcal{L}(\mathcal{X}^{(n)})$ yields*

$$\widetilde{\mathcal{L}}(t) \leq \mathcal{L}(\mathcal{X}^{(n)}) - \eta \|\nabla \mathcal{L}(\mathcal{X})\|_2^2 + \frac{M\eta^2}{2} \|\nabla \mathcal{L}(\mathcal{X})\|_2^2. \tag{22}$$

*Now we minimize over $\eta$ on both sides of (22), and denote the optimal value by $\widetilde{\mathcal{L}}(\eta^*)$. The right-hand side of (22) is simple quadratic, hence it is minimized by $\eta = 1/M$, and*

$$\min(\mathrm{RHS}) = \mathcal{L}(\mathcal{X}) - \frac{1}{2M} \|\nabla \mathcal{L}(\mathcal{X})\|_2^2. \tag{23}$$

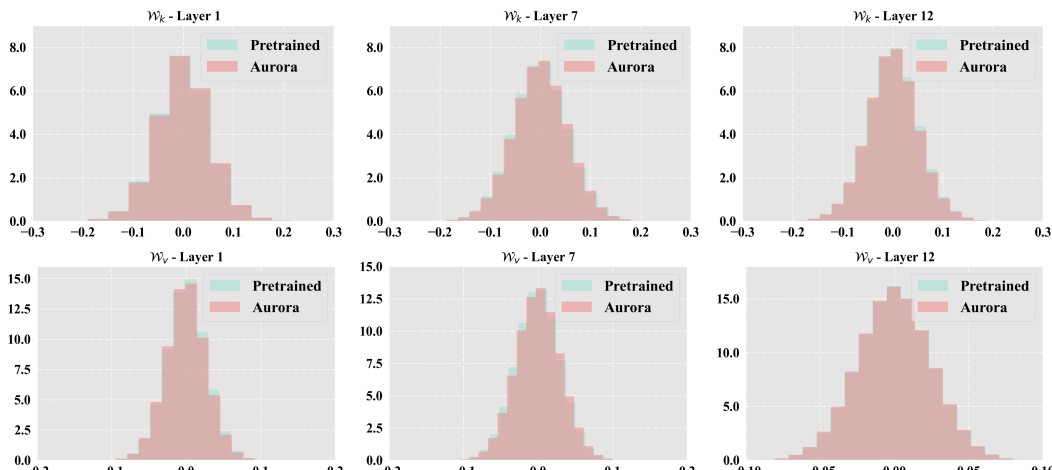

Figure 10: We represent the parameter distribution on different layers of the pre-trained large-scale multimodal foundation model (BLIP) *vs.* our Aurora, which is tuned on MSCOCO for image-text retrieval. Notably, $\mathcal{W}_k$ and $\mathcal{W}_v$ are the stack of the key and value projection matrices in different modality branches.

*Now,*

$$\mathcal{L}(\mathcal{X}^{(n)} + \eta\Delta\mathcal{X}^{(n)}) = \widetilde{\mathcal{L}}(t^*) \leq \mathcal{L}(\mathcal{X}^{(n)}) - \frac{1}{2M}\|\nabla\mathcal{L}(\mathcal{X})\|_2^2. \tag{24}$$

*Subtracting $\lambda^*$ on both sides, we have,*

$$\mathcal{L}\left(\mathcal{X}^{(n)} + \eta\Delta\mathcal{X}^{(n)}\right) - \lambda^* \leq \mathcal{L}(\mathcal{X}^{(n)}) - \lambda^* - \frac{1}{2M}\|\nabla\mathcal{L}(\mathcal{X})\|_2^2, \tag{25}$$

*by (20) we have,*

$$\mathcal{L}\left(\mathcal{X}^{(n+1)}\right) = \mathcal{L}\left(\mathcal{X}^{(n)} + \eta\Delta\mathcal{X}^{(n)}\right) - \lambda^* \leq \left(1 - \frac{m}{M}\right)\left(\mathcal{L}(\mathcal{X}^{(n)}) - \lambda^*\right). \tag{26}$$

*By mathematical induction, we obtain,*

$$\mathcal{L}(\mathcal{X}^{(n)}) - \lambda^* \leq \left(1 - \frac{m}{M}\right)^n \left(\mathcal{L}(\mathcal{X}^{(0)} - \lambda^*\right) \to 0 \quad (n \to \infty), \tag{27}$$

*therefore $\mathcal{L}(\mathcal{X}^{(n)}) \to \lambda^*$ as $n \to \infty$.*

*Suppose $\mathcal{V}^*$ is another optimal parameter tensor, then by the same argument we have $\mathcal{L}(\mathcal{X}^{(n)}) \to \mathcal{L}\left(\mathcal{V}^*\right)$ as $n \to \infty$. Since $\mathbb{R}^{ddN}$ is a Hausdorff space, $\mathcal{L}\left(\mathcal{V}^*\right) = \mathcal{L}\left(\mathcal{W}^*\right)$. By our assumption that $\mathcal{L}$ is injective, $\mathcal{W}^* = \mathcal{V}^*$.*

## G  Visualization Analysis

### G.1  Parameter Distribution

Figure 10 and Figure 11 show more details on the parameter distribution comparisons between our Aurora and the pre-trained model and the full fine-tuned model, in which similar results can be observed on $\mathcal{W}_k$ and $\mathcal{W}_v$. We can see that the mode approximation parameters adjust the original weights, and change the distribution of weights and biases to fit the downstream task. It can be concluded that Aurora has several advantages over traditional fine-tuning approaches. First, it avoids over-fitting to specific downstream tasks by only adjusting the pre-trained model parameters in a small local range. Second, it reduces the amount of training required on new data, making it more efficient and cost-effective. Last but not least, Aurora can further improve the model's performance on downstream tasks.

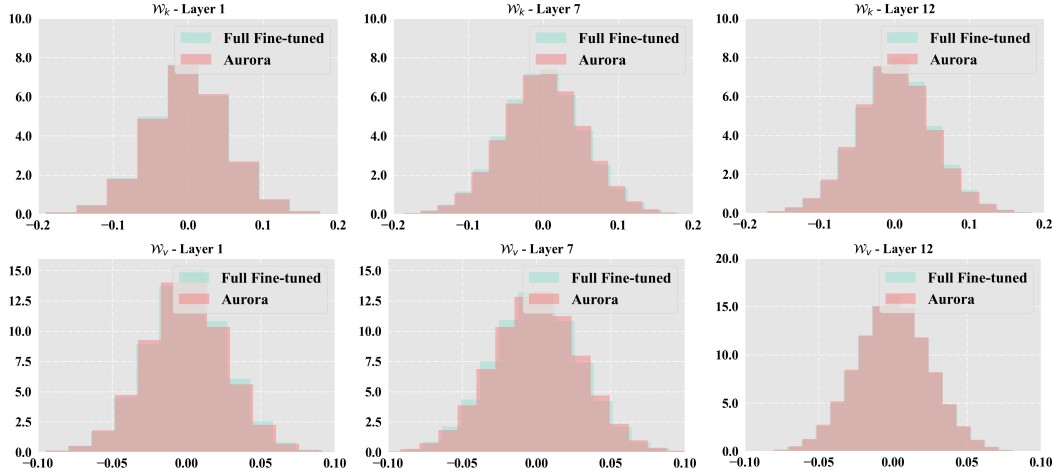

Figure 11: We represent the parameter distribution on different layers of the full fine-tuned model *vs.* our Aurora, which is tuned on MSCOCO for image-text retrieval. Notably, $\mathcal{W}_k$ and $\mathcal{W}_v$ are the stack of the key and value projection matrices in different modality branches.

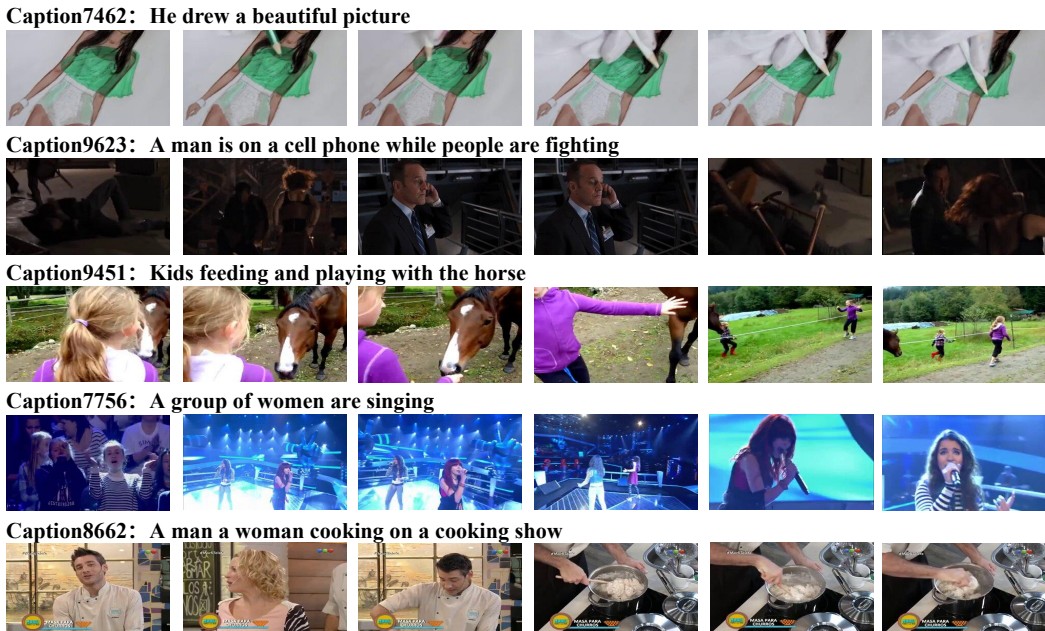

Figure 12: Video-Text retrieval cases on MSRVTT test set.

## G.2   Case Study

**Visual-Text Retrieval.** Figure 12 demonstrates some actual examples of Aurora performing text-to-video task on MSRVTT test set. In conclusion, the results presented highlight the exceptional performance of Aurora in searching relevant videos from textual descriptions. The accuracy and realism of the returned videos demonstrate the effectiveness of our proposed method in understanding the relationship between text and visual content.

**Visual Question Answering.** Figure 13 gives some question-answering examples of Aurora and UniAdapter on the MSRVTT-QA dataset. Specifically, our method is able to reason about the meaning of the text and video information to answer the questions more accurately than UniAdapter. This is an important result because the ability to reason about the meaning of both text and visual information is essential for understanding multimodal data.

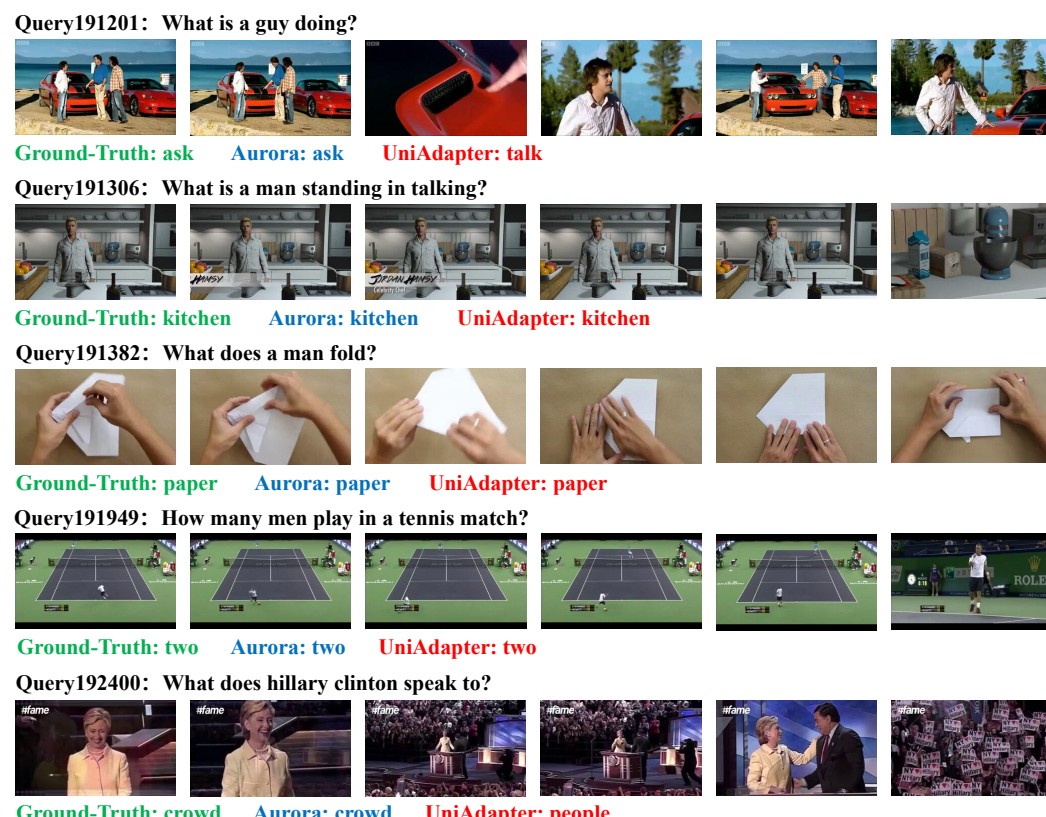

**Query191201: What is a guy doing?**

**Ground-Truth: ask**   **Aurora: ask**   **UniAdapter: talk**

**Query191306: What is a man standing in talking?**

**Ground-Truth: kitchen**   **Aurora: kitchen**   **UniAdapter: kitchen**

**Query191382: What does a man fold?**

**Ground-Truth: paper**   **Aurora: paper**   **UniAdapter: paper**

**Query191949: How many men play in a tennis match?**

**Ground-Truth: two**   **Aurora: two**   **UniAdapter: two**

**Query192400: What does hillary clinton speak to?**

**Ground-Truth: crowd**   **Aurora: crowd**   **UniAdapter: people**

Figure 13: Video Question Answering cases on MSRVTT-QA test set.

Overall, the qualitative results shown in Figure 12 and Figure 13 demonstrate the effectiveness of our proposed method in both multimodal retrieval and question-answering tasks. We believe that our approach has the potential to be used in many multimodal applications, where understanding and analyzing multimedia data is essential.

