# OpenReview forum: "Parameter-efficient Tuning of Large-scale Multimodal Foundation Model"
_NeurIPS.cc/2023/Conference — NeurIPS 2023 poster_

### Official Review · Reviewer_Srms · 2023-07-05

**Soundness:** 3 good
**Presentation:** 3 good
**Contribution:** 3 good
**Rating:** 5
**Confidence:** 4

**Summary:**

This paper proposes a novel approach to address the challenge of high learning costs when migrating large models to specific downstream tasks. The proposed method aims to reduce task complexity and improve the consistency across the different modal outputs of multimodal models.

The authors employ a LoRA-like technique that involves fine-tuning the transformer by appending adjustment matrices to the qkv matrices. This process is further optimized through CP Decomposition, an approach used to reduce the scale of the fine-tuning parameters. Then use Informative Context Enhancement mechanism to compute weights and the mixed graph-text features are adjusted according to these weights, facilitating the generation of superior fused features. To prevent the loss of textual information during the alignment of deep multimodal networks, the authors propose using a Gated Query Transformation. This approach calculates the blending ratio of text features for enhancement.

The paper claims that these techniques allow the model to outperform the current best methods on several downstream tasks, using fewer fine-tuning parameters, and even surpasses fully fine-tuned methods.

**Strengths:**

（1） Innovative and Effective Approach: Building upon the foundations of similar works like LoRA, this paper makes significant strides by achieving better fine-tuning performance with lower parameter usage.
（2） Proposes Novel Techniques: The paper introduces innovative methods such as Informative Context Enhancement and Gated Query Transformation to enhance the fusion of modalities. These methods appear to be highly effective in improving model performance, and could potentially be applied in a range of different contexts.
（3） High-Quality Visualizations: The paper includes aesthetically pleasing and informative figures and tables, which contribute to the clarity and overall quality of the work. The visualizations effectively aid in the understanding of complex concepts and methodologies.

**Weaknesses:**

(1)  Not Clearly Written: There appears to be some confusion in the terminology used, especially around 'soft prompts'. While the paper asserts that its fine-tuning approach can be seen as a 'soft prompt', the proposed method is more akin to parameter-efficient transfer learning. Therefore, the use of the term 'prompt' may not be accurate in this context. A clearer and more precise usage of technical terminology would be beneficial for readers and would also strengthen the overall quality of the paper. (Now author states that in the final version, they will replace the inappropriate term "prompt" with "multimodal parameter-efficient transfer learning based on mode approximations" to better reflect the essence of our method.)
(2) Lack of Details on Gated Query Transformation: The paper does not provide a clear and thorough explanation of the implementation details for the Gated Query Transformation. For future improvements, it would be beneficial to include more technical details of this novel technique, which would improve the clarity of the paper and make it easier for others in the field to replicate and build upon this work. (Now author states the final version will be more detailed)
(3) Manual Parameter Tuning: The paper suggests that the rank hyperparameter needs to be manually adjusted, which could pose an obstacle to scalability and efficiency. More research is required on how to optimally choose this hyperparameter's value. (Author said they use a detailed grid search experiments to get good rank)

**Questions:**

(1) Justification for Terminology: Could you clarify why you choose to refer to your method as a 'multimodal prompt'? Given the technique's resemblance to low-rank fine-tuning, the terminology might be misleading. Understanding the rationale behind this terminology would greatly aid in interpreting your work.
(2) Clarification on Gated Query Transformation: The paper could benefit from further details on the application of the Gated Query Transformation. Specifically, is the text information fused with the image features before entering the cross-attention, according to the gate values? This part of the methodology was not entirely clear in the paper, and further explanation would enhance the clarity of your methods and potentially support the broader understanding and application of your techniques.

**Limitations:**

(1) Hyperparameter Selection: The need for manual selection of the rank hyperparameter is a key limitation of the proposed method. While it is acknowledged by the authors, there is no substantial discussion on how this limitation can be addressed.
(2) Training Time: Another potential limitation that should be acknowledged is the impact on training time. Although the proposed method reduces the parameter count for downstream tasks, it doesn't seem to significantly decrease training time.

---

> ### Author Rebuttal · Authors · 2023-08-10
>
> Thank you for your recognition of the novelty and effectiveness of our approach as well as the visualization demonstration. We will continue to improve based on your feedback, and we believe that our Aurora has a very positive impact on promoting the efficient transfer of multimodal large models in the community.
>
> >**Not Clearly Written.** Thanks for the comment on the terminology and we greatly appreciate your valuable feedback. We agree with your statement, and our method is essentially a parameter-efficient transfer learning method. Our initial understanding is that the features obtained by passing the input through the learnable parameters constructed by mode approximation are prompts, which are added to features obtained by the pre-trained network. Therefore, in the final version, we will replace the inappropriate term "prompt" with "multimodal parameter-efficient transfer learning based on mode approximations" to better reflect the essence of our method.
>
> >**Lack of Details on Gated Query Transformation.**
> Thanks for your sincere comment on the detail supplement and we greatly appreciate your valuable feedback. We will add solid proof behind the argument regarding the loss of textual information and implementation details in our final version for better understanding.
>
> >**Manual Parameter Tuning.**
> Thanks for the comment on the hyper-parameter and we greatly appreciate your valuable feedback. Actually, we borrow the idea of [1] to automatically estimate the intrinsic dimension and then tune the model according to the estimated rank value (around 60). And then to further investigate the effectiveness of our low-rank decomposition, we implement a detailed grid search experiments on different rank values, which is shown in the Figure 3 in the paper. And an obvious phenomenon is that the performance improvement starts to level off after the rank increases to 64. Furthermore, we will attempt to intergrate the intrinsic dimension estimator with our method more tightly.
>
> >**Justification for Terminology.**
> Our initial understanding is that the features obtained by passing the input through the learnable parameters constructed by mode approximation are added to the features obtained by the pre-trained network as prompts. As a result, our mode approximation modules in the multi-modal network are called `multimodal prompt'. In the final version, we will replace the inappropriate term "prompt" with "multimodal parameter-efficient transfer learning based on mode approximations" to more accurately reflect the nature of our method.
>
> >**Clarification on Gated Query Transformation.**
> Thanks for your sincere comment on the detail supplement and we greatly appreciate your valuable feedback. Loss of textual information in deep multi-modal fusion branches essentially forms the basis of introducing Gated Query Transformation. Please refer to the answer for reviewer NuU2 for details of solid proof. Gated Query Transformation utilizes the gated function to fuse textual information with the feature in the fusion branch as the input for the cross-attention in the next layer to avoid textual information loss. We will add the solid proof behind the argument regarding the loss of textual information and implementation details in our final version for better understanding.
>
>
> [1] Chen B, Huang K, Raghupathi S, et al. Automated discovery of fundamental variables hidden in experimental data[J]. Nature Computational Science, 2022, 2(7): 433-442.

---

> > ### Comment · Reviewer_Srms · 2023-08-14
> >
> > Thanks your feedback, After a careful re-evaluation of your paper and its rebuttal, I acknowledge you addressed our initial concrens about "prompt". Your method will be a useful way to finetuning a VLM.

---

> > > ### Author Response · Authors · 2023-08-14
> > > **Response to reviewer Srms**
> > >
> > > Thank you once again for providing us with your valuable feedback on our paper. We are grateful to learn that our responses have successfully addressed your concerns. Your efforts in reviewing our work, as well as your insightful comments and support, are sincerely appreciated. Your suggestion has been invaluable in shaping the final version of our paper. We genuinely value your contributions and will ensure that your valuable suggestions are carefully incorporated.

---

> ### Author Response · Authors · 2023-08-17
> **Thanks for your efforts and look forward to your reply.**
>
> We sincerely appreciate your review and the constructive suggestions you have provided once again! Through our discussions and the reviewers' responses, it appears that we have effectively addressed the major concerns raised by everyone, and received a higher score from Reviewer kJXf. This outcome has greatly benefited us, and we would like to express our gratitude to all of you for your support!
>
> &emsp;
>
> After carefully reviewing your feedback once again, we have summarized the key points and will implement these modifications in the next version:
> * Rectify the use of "prompts" and replace it with "parameter-efficient transfer learning method" for accurate representation.
> * Supplement more details regarding the important modules and polish up the writing
> * Supplement broad details on our training like parameter tuning tricks.
>
> &emsp;
>
> We firmly believe that our framework (AURORA) for parameter-efficient transfer of multimodal models plays a significant role in advancing the community. And we are committed to making our complete code and training details publicly available.
> Moreover, we are eager to engage in further discussions with you to enhance our understanding of the domain and further improve the quality of the paper.
>
> &emsp;
>
> And we deeply appreciate that if you could reconsider the score accordingly. We are always willing to address any of your further concerns.

---

### Official Review · Reviewer_kJXf · 2023-07-06

**Soundness:** 3 good
**Presentation:** 3 good
**Contribution:** 3 good
**Rating:** 5
**Confidence:** 4

**Summary:**

This paper aims to design a lightweight prompt tuning method (i.e. Aurora) for cross-modal transfer. The main idea follows the observation by LoRA [15] that most of the features are redundant and a low-rank ∆W can be learned to adapt the features. Different from LoRA, they adopt CP decomposition [47] to decompose the learnable parameters into a series rank-one tensors. They also propose Informative Context Enhancement and Gated Query Transformation for better modality alignment. However, the connection between the two parts is not clear. Experiments show that Aurora performs better than LoRA and is at least comparable to full finetuning methods.

**Strengths:**

Strength
1.	Figure 1 highlights the difference between the proposed method and the baselines.
2.	The proposed method is lightweight and effective.
3.	Aurora is applicable to both image-text and video-text retrieval.


**Weaknesses:**

Major
1.	The main idea of the paper is lightweight adaptation achieved by adopting CP decomposition [47] for the adapter weights. While it is shown to be effective and parameter-efficient, there is limited technical novelty introduced by this work. In terms of story-level novelty, it mainly follows the observation and approach in LoRA [15], and it’s not considered novel.

2.	The improvement of Aurora over UniAdapter [32] is marginal in Table 1.

3.	What are BLIP, BLIP+LoRA zero-shot performance in Table 4?

4.	The new thing of this paper is Informative Context Enhancement and Gated Query Transformation. However, there is loose connection between the main idea and these two modules. These two modules are orthogonal to the low-rank approximation part. What will be the performance if LoRA is integrated with these two modules?

5.	Which part of the method does Parameter Sharing in L248-L255 correspond to?

Minor
6.	What are x-axis and y-axis in Figure 7?

7.	The main flow of the architecture is not clear. In section 3.3, Gated Query Transformation is presented after Context Enhancement. However, Gated Query Transformation manipulates f, but Context Enhancement depends on f. Is the former performed prior to the latter?

8.	The insight for Gated Query Transformation is unclear. How about replacing t’ in L191 with t?


**Questions:**

The author is suggested to address the concerns in the weakness section.

**Limitations:**

The limitation is mentioned at the end of the paper.

---

> ### Author Rebuttal · Authors · 2023-08-10
>
> Thank you for your recognition of the lightweight design, effectiveness, and comprehensive experiments of our method. We will continue to improve based on your feedback, and we believe that our Aurora has a very positive impact on promoting the efficient transfer of multimodal large models in the community.
>
> >**Limited technical novelty.** We believe that our paper has certain novelty and contributes to the entire community for three reasons. **First**, our main advantage is that the mode approximation based on CP decomposition has a lighter parameter decomposition architecture compared to LoRA. **Second**, our low-rank decomposition method has better mathematical interpretability as theoretical support (please refer to Appendix F). Most importantly, **third**, Aurora is not significantly dependent on rank, demonstrating true high parameter efficiency.
> Specifically, in Figure 3 of the main text, part (d), we can observe that as the rank increases, our method Aurora does not exponentially increase the burden of learnable parameters. Experimental results have further validate our idea.
>
> > **Marginal improvement in Table 1.** We would like to point out that our advantages beyond UniAdapter are actually clear not only on parameter efficiency but also on performance.
> **First**, in the cross-modal retrieval task in Table 1, we achieve better results than UniAdapter even when the fine-tuned accuracy is already near 100\%.
> **Second**, in the tasks of Table 2 and 3, we achieve even greater advantages over UniAdapter, with around a 3\% improvement in performance in both video-text retrieval and VQA tasks.
> **Third**, our Aurora already achieves a leading advantage with rank=64. In fact, when we compare fairly with UniAdapter using rank=512, our leading advantage will be further expanded to around 5\%.
>
> > **Zero-shot performance in Table 4.** We add additional experiments on BLIP and BLIP+LoRA, where BLIP is the base pretrained version and BLIP+LoRA is fintuned with LoRA. The experiment results in Table are shown below:
> | Method | \# Parameter ||   MSRVTT (T2V)  |   | | DiDemo (T2V)   |   |
> |-- | :------: | :--: | :--: | :--: | :--: | :--: | :--: |
> |    |    | R@1 | R@5  | R@10 | R@1 | R@5  | R@10 |
> | BLIP |  223M |  41.5 | 62.0 | 70.7 |42.1   | 59.6 | 67.3 |
> | BLIP + LoRA |  10.6M| 42.7  | 62.8 | 71.4 |  43.3   | 60.3 | 68.2 |
>
> >**Loose connection between main modules.** Mode approximation is designed for high-efficiency transfer, however, there exists no pure module that can be adapted and suffers from no modality alignment pain. Therefore, how to utilize the feature outputs of the mode approximation module to boost the performances on multi-modal tasks is the core motivation of Informative Context Enhancement and Gated Query Transformation. We also add some experiments on **LoRA integrated with Informative Context Enhancement and Gated Query Transformation**. The results are shown below, we can draw the following conclusions: **First**, Informative Context Enhancement and Gated Query Transformation indeed boost the performances of multi-modal tasks even with LoRA, which validates the effectiveness of our proposed module. **Second**, the increase on LoRA is obviously lower than that on Aurora, which can be attributed to that better representations learned on downstream tasks cause better modality alignment results.
> | \#Tunable |  | MSCOCO I2T |  |  | MSCOCO T2I |  |  | FLICKR30K I2T |  |  | FLICKR30K T2I |  |
> |:---:|:---:|:---:|:---:|:---:|:---:|:---:|:---:|:---:|:---:|:---:|:---:|:---:|
> | - | R@1 | R@5 | R@10 | R@1 | R@5 | R@10 | R@1 | R@5 | R@10 | R@1 | R@5 | R@10 |
> |  10.6M  | 80.1 | 94.4 | 97.5 | 62.3 | 84.5 | 90.9 | 96.5 | 99.9 | 100.0 | 86.2 | 97.4 | 98.7 |
> | **\#Tunable** |  | **MSRVTT** |  |  |  |  |  | **DiDemo** |  |  |  |  |
> | -  | R@1 | R@5 | R@10 | MdR |  |  | R@1 | R@5 | R@10 | MdR |  |  |
> | 10.6M  | 50.6 | 72.7 | 81.6 | 2.0 |  |  | 51.7 | 76.0 | 83.4 | 2.0 |  |  |
>
> > **Parameter Sharing explanation.** The parameter sharing described in L248-L255 means that the trainable parameters of the textual branch, visual branch, and multimodal fusing branch in BLIP share the same U and V decomposition factors when performing mode approximation. Detailed descriptions can be found in L144-L146.
>
> >**Figure 7 explanation.** Figure 7 shows a comparison of the distribution statistics of the parameters for the pre-trained model and our Aurora after efficient fine-tuning. The x-axis represents the parameter values, and the y-axis represents the frequency of occurrence.
>
> >**Demonstration order.** Gated Query Transformation is implemented prior to Context Enhancement following Figure 2. We will change the order in which these two modules are presented in the final version to help readers better understand.
>
> >**Insight for Gated Query Transformation.** Loss of textual information in deep multi-modal fusion branches essentially forms the basis of introducing Gated Query Transformation. The solid proof is also given in A2 for reviewer NuU2. Since $t'$ is learned by autograd, we output the mean value of the zero-initialized learnable transformation matrix $\gamma$ and $\beta$, which is 1.17, 0.23 (1.09, 0.36) on Didemo (Flickr30K) separately. It demonstrates that scaling textual information is beneficial for training.

---

> > ### Comment · Reviewer_kJXf · 2023-08-15
> >
> > Thanks the author for providing the detailed rebuttal. The additional experiments and explanation has addressed my concern and I would like to increase the score.

---

> > > ### Author Response · Authors · 2023-08-16
> > > **Response to Reviewer kJXf**
> > >
> > > Thank you once again for providing us with your valuable feedback on our paper. We are grateful to learn that our responses have successfully addressed your concerns. Your efforts in reviewing our work, as well as your insightful comments and support, are sincerely appreciated. We sincerely appreciate your willingness to increase the score based on these improvements. Once again, thank you for your valuable feedback and support.

---

> ### Author Response · Authors · 2023-08-17
> **Thanks for your efforts and look forward to your reply.**
>
> We sincerely appreciate your review and the constructive suggestions you have provided once again! Through our discussions and the reviewers' responses, it appears that we have effectively addressed the major concerns raised by everyone, and received a higher score from you. This outcome has greatly benefited us, and we would like to express our gratitude to all of you for your support!
>
> &emsp;
>
> After carefully reviewing your feedback once again, we have summarized the key points and will implement these modifications in the next version:
> * Enhance the clarity of our novelty in writing and provide a comprehensive explanation of the motivations behind crucial modules.
> * Supplement additional ablative experiments to further validate the effectiveness of our method and important modules.
> * Refine the paper's details, such as the writing flow, interpretation of figures and tables, to reduce confusion.
>
> &emsp;
>
> We firmly believe that our framework (AURORA) for parameter-efficient transfer of multimodal models plays a significant role in advancing the community. And we are committed to making our complete code and training details publicly available.
> Moreover, we are eager to engage in further discussions with you to enhance our understanding of the domain and further improve the quality of the paper.
>
> &emsp;
>
> We are always willing to address any of your further concerns.

---

### Official Review · Reviewer_ME39 · 2023-07-07

**Soundness:** 3 good
**Presentation:** 2 fair
**Contribution:** 3 good
**Rating:** 5
**Confidence:** 4

**Summary:**

This paper proposes a parameter efficient adaptation technique Aurora for multi-modal models. Particularly, the proposed method motivates their design by suggesting that the original pre-trained weight matrices have redundancies due to their high dimensional nature and the downstream task often requires low-dimensional reparameterization only. Aurora supplements the original weight matrices with series-one rank tensors which are only learned during the fine-tuning process.

In addition, to enhance the modality alignment between the vision and text representations, Aurora utilizes informative context enhancement module and gated query transformation which fuses and explicitly relates the textual representations with the multi-modal fusion representations in the cross-attention block of BLIP.

Extensive experiments over various benchmarks shows the effectiveness of Aurora in comparison with fine-tuning and parameter efficient adaptation approaches.

**Strengths:**

Strengths:

(1) The idea of decomposing learnable pre-trained matrices into small rank tensors is encouraging, as it explicitly allows to adapt only necessary amount of parameters for efficient and effective adaptation.

(2) The paper has performed extensive evaluations with proper ablation studies which justifies their design choices.

(3) The proposed methods performs favorably well with very less number of learnable parameters.

**Weaknesses:**

Weaknesses:

(1) The overall paper presentation style is very confusing, specially the main methodology section. There is no preliminaries on the baseline architecture on which the proposed solution has been built. It is very difficult for the readers to grasp the contents without knowing the main model architecture. For example, in line 161-162, the authors have mentioned the cross-attention module, but unfortunately no prior information about that block is provided anywhere in the manuscript. Also the writing is not clear and I found it difficult to understand the manuscript.

(2) How the proposed solution is considered as a prompt learning variant? If I understood correctly, the additional learnable parameters are utilized as part of the model and the input tensor has to be multiplied with it. It is not the case where the learnable parameters are part of the inputs, which is the core definition of prompt learning.

(3) The proposed multi-modal alignment module seems to be heavily designed for the BLIP multi-modal model. It is not clear if these components could be utilized in other multi-modal models. It will be good to see the generalization of proposed approach to other recent VL models.

**Questions:**

Please refer to the weaknesses section.

**Limitations:**

The authors have discussed the limitations and societal impacts are highlighted.

---

> ### Author Rebuttal · Authors · 2023-08-10
>
> Thank you for your recognition of our design idea and comprehensive experiments. We will continue to improve based on your feedback, and we believe that our Aurora has a very positive impact on promoting the efficient transfer of multimodal large models in the community.
>
> > **Confusing presentation.** Thanks for the comment on our work and we greatly appreciate your valuable feedback. We apologize for any confusion caused by the lack of preliminaries on the baseline architecture used to build our proposed solution. Due to the page limit, we put the details of the whole architecture of the pretrained model BLIP in Part D of the Appendix. Following your sincere suggestion to introduce more preliminary knowledge of our base model, we will revise the paper to include a more detailed explanation of the network architecture and its role in our proposed solution in a more clear way. We will work to improve the clarity of our writing to make it more accessible to readers in the final version.
>
> > **Wrong use of prompt learning.** Thanks for the comment on our work and we greatly appreciate your valuable feedback. Actually, prompt tuning is one typical way of parameter-efficient transfer learning (PETL). Our work is based on the existing roadmap of the PETL by decomposing the pre-trained networks with learnable parameters. These learnable parameters are multiplied with the input as the "soft prompts" for the pre-trained parameters to implement PETL. Therefore, our Aurora is quite different from the typical prompts as the inputs. In the final version, we will replace the "prompt" with "multimodal parameter-efficient transfer learning based on mode approximations" to better reflect the essence of our method.
>
> > **Generalization results.** Following your suggestion to further validate the generalization ability, we extend our Aurora to a more recent sota vision-language model, InstructBLIP [1]. We apply Aurora to the Q-former architecture in InstructBLIP, and the results are shown below.
> | Method              | OKVQA | A-OKVQA | COCO Caption |
> |----------|:--------:|:-----:|:----------:|
> | InstructBLIP+FTE（188M） | 54.9  | 55.9    | 68.0         |
> | InstructBLIP+LoRA（11M）        | 53.3  | 52.8    | 67.4         |
> | InstructBLIP+UniAdapter（18M）        | 53.2  | 53.5    | 67.2         |
> | InstructBLIP+Aurora（0.5M）      | 53.7  | 54.1    | 67.6         |
>
>
> [1] Liu H, Li C, Wu Q, et al. Visual instruction tuning[J]. arXiv preprint arXiv:2304.08485, 2023.

---

> > ### Comment · Reviewer_ME39 · 2023-08-15
> >
> > Dear Authors,
> >
> > Thank you for providing a rebuttal response, it has majorly addressed my concerns.
> >
> > Yes, the final version manuscript should use replace the "prompt learning" phrase to avoid any confusion in the research community.
> >
> > Based on the rebuttal response, I will keep my current score.

---

> > > ### Author Response · Authors · 2023-08-15
> > > **Response to  Reviewer ME39**
> > >
> > > Thank you for your response. We greatly appreciate your acknowledgment that our rebuttal has effectively addressed your concerns.
> > > We have duly noted your suggestion regarding the replacement of the term "prompt learning" in the final version of the manuscript. We will ensure that this change is made to avoid any potential confusion within the research community. Once again, we would like to express our gratitude for your valuable feedback and for contributing to the improvement of our manuscript.

---

> ### Author Response · Authors · 2023-08-17
> **Thanks for your efforts and look forward to your reply.**
>
> We sincerely appreciate your review and the constructive suggestions you have provided once again! Through our discussions and the reviewers' responses, it appears that we have effectively addressed the major concerns raised by everyone, and received a higher score from Reviewer kJXf. This outcome has greatly benefited us, and we would like to express our gratitude to all of you for your support!
>
> &emsp;
>
> After carefully reviewing your feedback once again, we have summarized the key points and will implement these modifications in the next version:
> * Add a subsection about ‘Revisiting Backbone’ to introduce the base model.
> * Rectify the use of "prompts" and replace it with "parameter-efficient transfer learning method" for accurate representation.
> * Conduct experiments on two additional base models to validate our advantages on the generalization.
>
> &emsp;
>
> We firmly believe that our framework (AURORA) for parameter-efficient transfer of multimodal models plays a significant role in advancing the community. And we are committed to making our complete code and training details publicly available.
> Moreover, we are eager to engage in further discussions with you to enhance our understanding of the domain and further improve the quality of the paper.
>
> &emsp;
>
> And we deeply appreciate that if you could reconsider the score accordingly. We are always willing to address any of your further concerns.

---

### Official Review · Reviewer_37bN · 2023-07-27

**Soundness:** 3 good
**Presentation:** 3 good
**Contribution:** 3 good
**Rating:** 5
**Confidence:** 1

**Summary:**

The paper addresses the problems of (i) transfer learning and (ii) reducing the multimodality gap in multimodal models.

To address (i) it presents a technique which can be viewed as a generalization of LoRA; instead of independently factoring representing matrices as a low rank representation, all the matrices of the transformer get stacked together and are represented as a low rank tensor.

To address (ii) two techniques are presented: one which aims to improve the representations by allowing information exchange between different examples in the batch (Informative Context Enhancement) and another which aims to prevent the loss of text information for deep models (Gated Query Transformation).

Ablation studies are performed to justify the different design choices.


**Strengths:**

1. Mode approximation (Generalizing LoRA by stacking the matrices of all the transformer layers and using CP decomposition to represent that) is a nice idea and experimentally shows to be a parameter efficient way (beats LoRA) of adapting frozen models to new domains

2. Thorough ablation studies are performed to demonstrate the impact and justify existence of each presented component in the final model.


**Weaknesses:**

1. Informative Context Enhancement seems to allow information exchange between different examples in the batch. That makes it dependent on the batch size, but the impact of changing it is not evaluated.

**Questions:**

1. Why are some methods present in ‘Methods with frozen backbone’ in Table 2 omitted from the same section in Table 3? (e.g. LoRA)
2. Is |B| in the ‘Informative Context Enhancement’ section referring to the number of tokens in the entire batch?


**Limitations:**

The authors have adequately addressed the limitations.

---

> ### Author Rebuttal · Authors · 2023-08-10
>
> Thank you for your highly accurate summary of our work and recognition of our comprehensive experiments. We will continue to improve based on your feedback, and we believe that our Aurora has a very positive impact on promoting the efficient transfer of multimodal large models in the community.
>
> >**Impact of the batch size.** Thanks for the comment on our work and we greatly appreciate your valuable feedback. We implement different numbers of batch size to investigate the effectiveness of Informative Context Enhancement. Some results are shown below:
> >
> >| DataSet                          | Batch Size = 4 | Batch Size = 8 | Batch Size = 16 | Batch Size = 32 |
> >| -------------------------------- | :------------: | :------------: | :-------------: | :-------------: |
> >| MSCOCO (I$\rightarrow$T, R@1)    |      80.4      |      80.6      |      80.7       |      80.8       |
> >| Flickr30K (I$\rightarrow$T, R@1) |      96.9      |      97.1      |      97.2       |      97.2       |
> >| DiDemo (R@1)                     |      53.1      |      53.1      |      53.2       |      53.4       |
> >| MSRVTT-QA                        |      44.4      |      44.7      |      44.8       |      45.0       |
>
> >**Lack of results.** Thanks for the comment on our work and we greatly appreciate your valuable feedback. We apologize for the omission of the comparison experiment results on LoRA in Table 3. We have added them, and the complete results for Table 3 are shown below:
> >
> >| Method                       | #Tunable | test-dev  test-std | Method         | #Tunable | test acc |
> >| -------------------------------- | :----------: | :--------------------: | ------------------ | :----------: | :----------: |
> >| Methods with frozen backbone |              |                        |                    |              |              |
> >| LoRA (r=32)                      |    10.6M     |      74.11   74.24      | LoRA (r=32) |    10.6M     |     44.3     |
> |UniAdapter (r=512) | 18.8M | 75.44 75.56 | UniAdapter (r=512)| 18.8M | 44.7|
> |Aurora (r=64) | 0.1M | 77.69 77.87 | Aurora (r=64) | 0.1M | 44.8|
>
> >**Symbol not clear.** Thanks for the comment on our work and we greatly appreciate your valuable feedback. $| \mathcal{B}|$ is the number of image-text pairs in the entire batch, and "feature" means the [cls] token of each image and text. In other words, each [cls] token represents the global information of the image and text.

---

> > ### Comment · Reviewer_37bN · 2023-08-14
> >
> > Thank you for the thorough clarification of all my questions.

---

> > > ### Author Response · Authors · 2023-08-15
> > > **Response to Reviewer 37bN**
> > >
> > > Thank you for providing a comprehensive clarification of all my queries. Your detailed responses have been immensely helpful in enhancing my understanding of the topic. I genuinely appreciate the time and effort you have dedicated to addressing my concerns.

---

> ### Author Response · Authors · 2023-08-17
> **Thanks for your efforts and look forward to your reply.**
>
> We sincerely appreciate your review and the constructive suggestions you have provided once again! Through our discussions and the reviewers' responses, it appears that we have effectively addressed the major concerns raised by everyone, and received a higher score from Reviewer kJXf. This outcome has greatly benefited us, and we would like to express our gratitude to all of you for your support!
>
> &emsp;
>
> After carefully reviewing your feedback once again, we have summarized the key points and will implement these modifications in the next version:
> * Supplement more details regarding the ablation experiments for validation.
> * Provide further experimental writing details to better elucidate AURORA.
>
> &emsp;
>
> We firmly believe that our framework (AURORA) for parameter-efficient transfer of multimodal models plays a significant role in advancing the community. And we are committed to making our complete code and training details publicly available.
> Moreover, we are eager to engage in further discussions with you to enhance our understanding of the domain and further improve the quality of the paper.
>
> &emsp;
>
> And we deeply appreciate that if you could reconsider the score accordingly. We are always willing to address any of your further concerns.

---

### Official Review · Reviewer_NuU2 · 2023-07-27

**Soundness:** 3 good
**Presentation:** 3 good
**Contribution:** 3 good
**Rating:** 6
**Confidence:** 4

**Summary:**

The paper proposes AURORA, a method which uses mode approximation to boost the knowledge transfer in Vision Language models and enhances alignment between the modalities in a lightweight parameter efficient manner. In addition to this, the paper further proposes Context Enhancement module and  Gated Query Transformation module to boost the modality fusion in an adaptively controllable way. The proposed method is evaluated on six cross-modal tasks and two zero-shot tasks and is compared to existing PETL methods.

**Strengths:**

- The paper seems to be the first which has used mode approximation in Vision-Language models to efficiently achieve modality fusion.
- The proposed method is novel with respect to existing prompt tuning methods.
- The proposed methods works by tuning a very small number of parameters which can save time and computational resources.

**Weaknesses:**

- The motivation for using Mode approximation is not very clear in the paper. For instance there could be other methods which can solve the redundancies in the attention weights. There is no detailed explanation of the theoretical basis or formal analysis of how mode approximation aids in prompt learning. Adding a brief theoretical background could provide more insights into the method's underlying principles.
- The authors haven't provided any solid proof behind the argument regarding loss of textual information in multi-modality fusion branches, which essentially forms the basis of introducing Gated Query Transformation.
- The authors should have compared their method with previous prompt learning methods [1, 2] for zero-shot out-of-distribution classification tasks.


[1] M. U. khattak, H. Rasheed, M. Maaz, S. Khan, and F. S. Khan, “Maple: Multi-modal prompt learning”. IEEE/CVF Conference on Computer Vision and Pattern Recognition, 2023

[2] K. Zhou, J. Yang, C. C. Loy, and Z. Liu, “Learning to prompt for vision-language models”. International Journal of Computer Vision (IJCV), 2022

**Questions:**

- Can the method be applied to a broader range of vision-language tasks, and how well does it adapt to different modalities or data distributions?

**Limitations:**

- Please have a look at Questions and Weaknesses

---

> ### Author Rebuttal · Authors · 2023-08-10
>
> Thank you for your professional comments. We will continue to improve based on your feedback, and we believe that our Aurora has a very positive impact on promoting the efficient transfer of multimodal large models in the community.
>
> > **Motivation not clear.** The motivation for our mode approximation module lies in two folds. **First**, it can approximate the weights in large-scale model with a little learnable parameters. **Second**, it is based on the algorithm of CP decomposition which has good theoretical convergence property. We agree with you that theoretical background could provide more insights into the method. Therefore, we have provided a detailed concept definition in Appendix Part A . **Third**, we provide detailed theoretical analysis and derivation in Appendix Part F to validate that the proposed method can achieve good convergence.\
> We acknowledge that there are other methods to address redundancy in attention weights. However, Aurora employs mode approximation to generate lightweight prompts, which is an effective approach that achieves good performance with very few parameters. We believe the advantage of this approach is that it enables efficient parameter transfer with multimodal prompt tuning under extremely few parameters while reducing computational and storage costs while maintaining performance. The advantages of Aurora are further demonstrated in the experimental results.
>
> > **No solid proof.** We provide the motivation for designing the Gated Query Transformation module in the following text, which is that the information content of textual information decreases as the layer depth of cross-attention increases.\
> Specifically, in the cross-attention mechanism, the encoding vectors are used to calculate the attention distribution for generating the output vectors. And the query vectors are used to calculate the weights of the attention distribution, where the query vectors come from the textual tokens under the multimodal setting. \
> Suppose we have $L$ layers in the cross-attention mechanism, and the **encoding vector (from visual modality) for layer $l$ is denoted as $e_l$**, and the **query vector (from textual modality) for layer $l$ is denoted as $q_l$**. We aim to prove that with the increase of the layer number $L$, the textual information content of the query vector $q_L$, which is the encoding vector, becomes lower and lower.\
> We can use the concept of entropy to measure the information content of the query vector.
> Then we can view the query vector $q_L$ as a random variable that has all possible encoding vectors $e_l$ as its possible values. To compute the entropy of the query vector $q_L$, we need to calculate the probability distribution $p(q_L)$. We can assume that $p(q_L|e_1, ..., e_L)$ is the probability distribution of the query vector $q_L$ given all the $L$ encoding vectors. Then, we can use Bayes' theorem to transform $p(q_L|e_1, ..., e_L)$ into the product form of $p(e_1, ..., e_L|q_L)$ and $p(q_L)$ as follows:\
> $$ p(q_L|e_1, ..., e_L) = \frac{p(e_1, ..., e_L|q_L) p(q_L)}{p(e_1, ..., e_L)} $$\
> Since $p(e_1, ..., e_L)$ does not depend on $q_L$, we can treat it as a constant and get:\
> $$ p(q_L) = \frac{p(e_1, ..., e_L|q_L) p(q_L)}{p(e_1, ..., e_L)} $$\
> Then, we can express the entropy of the query vector $q_L$ as:\
> $$H(q_L) = -\sum_{e_1, ..., e_L} p(e_1, ..., e_L) \sum_{q_L} p(q_L|e_1, ..., e_L) \log p(q_L|e_1, ..., e_L) \
> = -\sum_{e_1, ..., e_L} p(e_1, ..., e_L) \sum_{q_L} p(q_L) \log p(q_L) \
> \sum_{e_1, ..., e_L} p(e_1, ..., e_L) \sum_{q_L} p(e_1, ..., e_L|q_L) \log p(e_1, ..., e_L|q_L)$$
> The first summation term represents the entropy of the marginal distribution of $q_L$, which is independent of $e_1, ..., e_L$. Therefore, we can treat it as a constant and rewrite the second summation term as: $ -\sum_{e_1, ..., e_L} p(e_1, ..., e_L) H(e_1, ..., e_L|q_L) $.\
> Here, $H(e_1, ..., e_L|q_L)$ is the conditional entropy of $e_1, ..., e_L$ given $q_L$. It represents the uncertainty of all possible encoding vectors $e_1, ..., e_L$ given the query vector $q_L$. Since the conditional entropy increases with the uncertainty of the conditioning variable, we can infer that the information content of $q_L$ decreases with the increase of $L$, as the uncertainty of the encoding vectors given $q_L$ increases.\
> In summary, as the layer number $L$ increases in the Transformer, the information content of the query vector $q_L$ decreases, indicating that the query vector becomes less informative about the encoding vectors in the subsequent layer.
>
> >**Lack of comparison.** Our paper mainly copes with the typical multi-modal tasks based on the BLIP architecture including cross-modal retrieval, VQA on both image and video modalities. We will further add the comparisons with the work you mentioned in the final version. During the rebuttal period, we would like to first add parts of the results which are shown below.
> | Datasets | ImageNet |       | Caltech101 |       | StanfordCars |       |
> |----------|:--------:|:-----:|:----------:|:-----:|:------------:|:-----:|
> | Method   |   base   | novel |  base  | novel |   base  | novel |
> | MaPLe    |   76.66  | 70.54 |  97.74 | 94.36 |  72.94 | 74.00 |
> | Aurora   |   76.59  | 70.75 |    98.13   | 94.52 |  73.75  | 74.28 |
>
> > **Generalization results.** Following your suggestion to further validate the generalization ability, we extend our Aurora to a more recent sota vision-language model, InstructBLIP[1]. We apply Aurora to the Q-former architecture in InstructBLIP, and the results are shown below.
> | Method              | OKVQA | A-OKVQA | COCO Caption |
> |----------|:--------:|:-----:|:----------:|
> | InstructBLIP+FFT（188M） | 54.9  | 55.9   | 68.0  |
> | InstructBLIP+LoRA（11M | 53.3  | 52.8    | 67.4  |
> | InstructBLIP+UniAdapter（18M） | 53.0  | 53.5    | 67.2 |
> | InstructBLIP+Aurora（0.5M) | 53.7  | 54.1    | 67.6  |
>
> [1] Liu H, Li C, Wu Q, et al. Visual instruction tuning[J]. arXiv preprint arXiv:2304.08485, 2023.

---

> > ### Comment · Reviewer_NuU2 · 2023-08-14
> >
> > Thank you for providing a detailed response to my queries.
> >
> > 1. I am satisfied with the Motivation statement you have provided. Indeed the theoretical proof is in agreement with your motivation.
> > 2. Regarding Gated Query Transformation, the entropy explanation seems fair and has a strong link with the motivation behind introducing Gated Query Transformation module.
> > 3.  For zero-shot out-of-distribution comparison, the results of AURORA seem to be close to MaPLe. However, given the scope of AURORA, the comparisons seem acceptable.
> > 4. Given very less parameters of AURORA, the generalization results are interesting. AURORA may prove to be a good low cost generalizable model.

---

> > > ### Author Response · Authors · 2023-08-14
> > > **Response to NuU2**
> > >
> > > Thank you once again for providing us with your valuable feedback on our paper. We are grateful to learn that our responses have successfully addressed your concerns. Your efforts in reviewing our work, as well as your insightful comments and support, are sincerely appreciated.
> > >
> > > We are pleased to receive your positive recognition of our experimental results. In the future, we will continue to analyze and enhance the performances on more base models (i.e., in MaPLE) to further amplify the advantages of our approach across a wider range of tasks. Moreover, we will make the code for our paper publicly available to facilitate research by a broader audience and foster advancements in the field. We really appreciate your efforts on reviewing our paper, your insightful comments and support.

---

> ### Author Response · Authors · 2023-08-17
> **Thanks for your efforts and look forward to your reply.**
>
> We sincerely appreciate your review and the constructive suggestions you have provided once again! Through our discussions and the reviewers' responses, it appears that we have effectively addressed the major concerns raised by everyone, and received a higher score from Reviewer kJXf. This outcome has greatly benefited us, and we would like to express our gratitude to all of you for your support!
>
> &emsp;
>
> After carefully reviewing your feedback once again, we have summarized the key points and will implement these modifications in the next version:
> * Highlight the motivation behind our method and important modules more prominently in the next version.
> * Provide additional detailed theoretical support in the appendix section.
> * Conduct experiments on two additional base models to validate our advantages on the generalization.
>
> &emsp;
>
> We firmly believe that our framework (AURORA) for parameter-efficient transfer of multimodal models plays a significant role in advancing the community. And we are committed to making our complete code and training details publicly available.
> Moreover, we are eager to engage in further discussions with you to enhance our understanding of the domain and further improve the quality of the paper.
>
> &emsp;
>
> And we deeply appreciate that if you could reconsider the score accordingly. We are always willing to address any of your further concerns.

---

### Decision · Program_Chairs · 2023-09-21

**Decision:**

Accept (poster)

**Comment:**

This is a borderline submission. The rebuttal and the following discussion clarified the remaining concerns and we believe that the propose approach achieves solid empirical results and provides sufficient insight to the relevant community. We urge the authors to incorporate the results from the rebuttal and discussion.